# Advances in the Use of Conducting Polymers for Healthcare Monitoring

**DOI:** 10.3390/ijms25031564

**Published:** 2024-01-26

**Authors:** Cuong Van Le, Hyeonseok Yoon

**Affiliations:** 1School of Polymer Science and Engineering, Chonnam National University, 77 Yongbong-ro, Buk-gu, Gwangju 61186, Republic of Korea; le.van.cuong.a@gmail.com; 2Department of Polymer Engineering, Graduate School, Chonnam National University, 77 Yongbong-ro, Buk-gu, Gwangju 61186, Republic of Korea

**Keywords:** conducting polymers, sensor technology, human health monitoring

## Abstract

Conducting polymers (CPs) are an innovative class of materials recognized for their high flexibility and biocompatibility, making them an ideal choice for health monitoring applications that require flexibility. They are active in their design. Advances in fabrication technology allow the incorporation of CPs at various levels, by combining diverse CPs monomers with metal particles, 2D materials, carbon nanomaterials, and copolymers through the process of polymerization and mixing. This method produces materials with unique physicochemical properties and is highly customizable. In particular, the development of CPs with expanded surface area and high conductivity has significantly improved the performance of the sensors, providing high sensitivity and flexibility and expanding the range of available options. However, due to the morphological diversity of new materials and thus the variety of characteristics that can be synthesized by combining CPs and other types of functionalities, choosing the right combination for a sensor application is difficult but becomes important. This review focuses on classifying the role of CP and highlights recent advances in sensor design, especially in the field of healthcare monitoring. It also synthesizes the sensing mechanisms and evaluates the performance of CPs on electrochemical surfaces and in the sensor design. Furthermore, the applications that can be revolutionized by CPs will be discussed in detail.

## 1. Introduction

Each day, the human body receives an immense amount of information, comprising both external stimuli and internally generated signals. The processing of these signals is integral to issuing commands that foster seamless coordination among various organs [1]. This coordination mitigates the adverse health effects induced by toxic chemicals. Sensory receptors located in the nose, eyes, ears, skin, and specific regions of the body predominantly capture these stimulating signals. Specifically, the human body possesses mechanisms to interpret these signals. Nature has endowed humans with remarkable mechanisms to capture and convey these stimuli. Nevertheless, in the context of modern industrial living, there exists a constant potential risk of events that can negatively impact human health, such as environmental pollution, exposure to chemical by-products leading to genetic mutations, or even disparities in individual operating time zones. The aforementioned challenges are not easily discernible by the body, as they manifest through subtle or undetectable signals that may not effectively communicate appropriate commands or alert about health risks [2]. This challenge is a primary motivator for advancing the development of more precise and efficient sensor designs. Considering the unique biological characteristics and constant activity of the human body, materials used for sensors should exhibit certain properties, specifically high flexibility and high biocompatibility [3].

In biosensing, diverse technologies address specific diagnostic needs. Continuous Glucose Monitoring Systems like Dexcom G6 and FreeStyle Libre transform diabetes management with real-time glucose data. Polymerase Chain Reaction-based biosensors, crucial in COVID-19 testing kits, amplify and identify viral genetic material for DNA detection. Lactate biosensors, such as the Lactate Plus Meter, offer insights into physical performance in sports and medical applications. Neurotransmitter biosensors assist neuroscience research by measuring neurotransmitter release for the study of neural communication. Handheld cholesterol testing devices empower individuals to monitor cholesterol levels at home for quick results. Troponin biosensors integrated into cardiac testing devices assess heart health by detecting troponin levels, indicating potential cardiac damage. Each biosensor type addresses specific diagnostic challenges, contributing to healthcare and safety. In recent years, conducting polymer (CPs) material has found extensive use across diverse fields, including solar cells [4], fuel cells [5], environment remediation [6,7], and biomedicine [8], because of the flexible nature of polymers and their ability to conduct electricity like metals. Simultaneously, the surge in biosensor applications has spurred a new direction in the application of CPs [9,10]. CPs offer unique advantages in biosensing applications. Many studies have developed CPs with outstanding physical and chemical properties, such as a large surface area [11], high electrical conductivity [12], flexibility [13], and stable mechanical properties [14,15]. In addition, CPs possess great polymer characteristics, such as ease of surface modification, ease of shaping the material as desired, and high biocompatibility. Therefore, CPs are excellent candidates for use in sensor contact interfaces [16]. Accordingly, many CP-based sensor designs have been successfully developed, including biochips [17], field-effect transistor biosensors [18], thermoelectric sensors [19], tactile sensors [20], strain sensors [21], piezoresistive sensors [22], bacteria sensors [23], gas sensors [24,25,26], cancer sensors [27], DNA chips [28], environment sensors [29,30,31], immune sensors [32], and pressure sensors [33], and they have demonstrated high-performance sensing properties. Figure 1 depicts CPs in the signal recognition of a sensor design.

However, there is room for improving CPs, specifically in terms of their detection limit, response speed, operating temperature range, and selectivity [34]. Furthermore, the expected surge in computing power and wearable electronic devices requires a corresponding enhancement in the ability to provide both qualitative and quantitative sensor inputs [35]. Notably, CPs are emerging as pivotal materials for biosensors, and they are likely to attract investment in various fields. Some well-known CPs, including polyacetylene [36], polydiacetylenes [37], polyaniline [38], polypyrrole [39], polythiophene [40], poly(3,4-ethylene-dioxythiophene) [41], poly(phenylene vinylene) [42], poly(3-hexylthiophene-2,5-diy [43], polyindole [44], poly(p-phenylene) [45], poly(3-alkylthiophene) [46], poly(p-phenylene-terephthalamide) [47], poly(isothianaphthene), poly(α-naphthylamine), polyazulene [48], polyfuran, polyisoprene [49], polybutadiene [50], poly(3-octylthiophnene-3-methylthiophene) [51], polyorthotoluidiene [52], poly (dioctylfluorene) [53], poly(p-phenyleneethynylene) [54], polyphenylene sulfide [55], and poly(triaryl amine) [56], have found diverse applications in the physicochemical interface. Figure 2 illustrates 20 popular types of CPs used in sensors.

CPs applications span a wide spectrum of uses, including rechargeable batteries [57], supercapacitors [58,59], electromagnetic shields [2], photocatalysts [60], anticorrosion coatings [61], and 3D printing technology [62,63]. CPs have played a significant role in advancing technology and enabling innovative solutions in these areas. Allotropes and nanomaterials used in combination with and to modify CPs exhibit significant diversity and considerable application potential. They are incorporated in zero-dimensional nanomaterials such as fullerenes [64], quantum dots [65,66,67,68,69], metal and metal oxides [70,71,72]; one-dimensional nanomaterials such as carbon nanotubes [73], silver nanowires [74] and titanium dioxide nanorods [75]; two-dimensional nanomaterial such as boron nitride nanosheets [76], hydrate vanadium dioxide nanosheets [77,78], and graphene [79,80]; and three-dimensional bulk materials such as silicon semiconductors [81], barium titanate filler [82], graphene foam [83], and indium gallium zinc oxide amorphous semiconductors [84]. Consequently, CPs have garnered substantial research interest, with research progressing in two primary directions: improving synthesis methods for the development of CP structures and identifying materials with which CPs can be combined to yield unique properties. However, the multitude of options related to the precursor materials used with CPs, their proportions relative to CPs, and diverse synthesis methods make it challenging to determine the potential applications of different combinations of CPs and precursors. This challenge is particularly pronounced in precision-dependent fields such as sensor technology [85,86].

In addition, different fields require specific levels of sophistication and sensitivity in sensors, a parameter significantly influenced by the conductivity of the CP at the contact surface. Notably, disparities in the application of preparation methods across CP studies can lead to research gaps. Despite many articles exploring the classification, manufacturing, applications, and sensing mechanisms of CPs, our focus is on the effect of CP properties on their effectiveness and use in collecting tracking data for monitoring the human body and health. This review follows a structured bottom-up approach. First, we present an overview of the role of CPs in sensor performance and examine the effect of CP properties on sensor performance. Second, we delve into advanced sensing materials, highlighting the combination of CPs with various inorganic hybrid materials such as nanocarbons, metal nanoparticles, and metal oxides. In addition, we discuss multicomponent composite CP systems synthesized through polymerization techniques, highlighting the growing trend of combining diverse CP structures for enhanced functionality in sensing applications. This discussion aims to show how the combinations contribute to improved sensor performance and an overall enhancement of the properties of CP materials. Third, we assess recent advances made in leveraging the unique characteristics of CPs in the design of purpose-specific sensors for human health monitoring and surveillance. Finally, we conclude the review by discussing the current challenges in the use of CPs for sensors and the future prospects of CPs in sensor technology.

## 2. Overview of the Use of CP in Sensor Design

### 2.1. CP Characterization

Researchers are drawn to CPs because of their controllable wide range of electrical conductivity and flexible mechanical properties. The alternating single-bond and double-bond structure (π-conjugated backbone) and the controllability of the type and level of doping are particularly advantageous. Figure 3 shows the conduction mechanism of PAc, which was the first synthesized CP based on π-conjugated backbone extension, allowing electron movement within the conjugated polymer chain [87]. Achieving high conductivity in CPs requires a structure with overlapping p-molecular orbitals and a high degree of π-bond conjugation. CPs feature an extensive π-conjugated system characterized by a network of irregular single and double bonds along the polymer chain. In their neutral state, highly conjugated CPs function as insulators. However, conductivity is achieved only when a p-bond electron is extracted from the π-conjugated polymer backbone, which creates a positively charged defect known as a polaron. The electrical conductivity of the CP–conjugated chain is affected by the type of doping and CPs electronic transitions that occur because of the presence of impurities, which alter the amount of charge within the band gap. The doping process in CPs differs considerably from that in inorganic semiconductors such as silicon and germanium in terms of the doping concentration and charge transformation of the conduction band energy. Doping in CPs involves the partial oxidation or reduction of the polymer, in which reactive ions are introduced to maintain electrical neutrality. This can be achieved either chemically or electrochemically by introducing cations or anions to balance the charge of the conjugated polymer through oxidation (p-doping) or reduction (n-doping) [88]. The degree of doping in CPs is significantly higher than that in traditional semiconductors, with approximately one-third to two-thirds of the monomer units being doped, which corresponds to doping concentrations falling within the range of 10^18^ to 10^22^ cm^−3^. The doping level is the main difference between CPs and other semiconductor materials. CPs exhibit distinct electronic properties that distinguish them from inorganic crystalline semiconductors in two critical aspects: their long-range order and molecular nature. The movement of charge carriers is the primary mechanism of electrical conduction in doped CPs. During the doping process, charges are either added to or removed from the polymer, resulting in the generation of mobile charge carriers capable of moving along the conjugated polymer chains because of the rearrangement of double and single bonds in the conjugated system [8]. This process transforms the insulating polymer into a conductive material. The extraction of p-bond electrons induces the delocalization of the remaining electrons within the p-orbitals along the length of the π-conjugated backbone, enabling the unrestricted movement of charge along the chain. Doping alters the bandgap energy of the polymer, leading to the increased conductivity of CPs [89,90].

The simplest CP structure comprises a backbond series of π-conjugated polymers belonging to polyenes such as PAc and PIP. The more complex structures are π-conjugated polymers belonging to polyaromatics such as PANi, PPy, PTh, PPP, and PPV classes, and they have been extensively studied. Another type of CP is a copolymer of a CP and another polymer, such as PEDOT:PSS [91], PANi:poly(methyl methacrylate) (PMMA) [92], PPy:poly(dimethyl siloxane) (PDMS) [93], PPy:poly(*N*-vinylcarbazole) [94], and P3HT:poly(ethylene glycol) [95]. Polymers featuring amino groups (-NH_2_) and carboxyl groups (-COOH) along the CP backbone readily facilitate bioconjugation with essential biorecognition molecules, such as enzymes, antibodies, proteins, and DNA [96]. Thiolate bio molecules, particularly mRNA, can autonomously assemble on CPs decorated with silver and gold nanoparticles. The fusion of CPs with water-swollen hydrogels yields porous CP hydrogels that provide a plethora of advantageous characteristics for biosensing platforms. These porous CP hydrogels exhibit exceptional electrical conductivity, robust mechanical properties, a sophisticated hierarchical structure, and the ability to absorb and release substances because of the inherent swelling capability of the polymer chain [97].

### 2.2. CP Processing

Achieving the required sensitivity of a sensor requires adjusting its properties before its use. The precursor material and manufacturing method determine most of the sensor’s properties. For CP materials and the synthesis environment, three main methods are employed: chemical oxidate polymerization, vapor-phase polymerization, and electrochemical polymerization, which lead to the formation of structures containing π-conjugates [98]. An alternative chemical method called vapor-phase polymerization has gained significant attention because of its ability to produce CP films with different thicknesses, uniformities, and densities. Typically, a CP is synthesized directly on a substrate surface through a two-step process. In the first step, an oxidizer, such as iron thiocyanate, is applied to the surface through a solvent coating process. Subsequently, the surface is exposed to CP monomer vapor to induce the polymerization reaction [99]. This method is commonly employed to synthesize PEDOT and PANi for their application in electrodes. The three main CP polymerization methods are shown in Figure 4.

#### 2.2.1. Chemical Oxidate Polymerization

Chemical oxidation polymerization is commonly used to synthesize PPy, PANi, PTh, and PIN. This method is based on redox principles and stimuli such as optical, physical, and biological stimuli [100]. The monomers used in this method exhibit electron-donating properties and a high tendency to undergo oxidation. Monomer oxidation is achieved using different oxidizing agents that generate cationic radical sites in the monomer [101]. Chemical polymerization can involve two mechanisms: polycondensation and chain growth. Polycondensation involves the recombination of cation radical oxidation sites, whereas chain growth is characterized by electrophilic substitution. The chemical oxidative method for CP polymerization has several advantages: it is suitable for producing large high-quality polymers, it allows the use of inexpensive and diverse oxidizing agents, and the resulting polymers exhibit stability with significant conductivity. A conductivity of 9.1 S/cm and an 80% increase in conductivity were achieved for PTh prepared by chemical oxidation with FeCl_3_ as the oxidant. The conductivity increased up to an oxidant–monomer molar ratio of 1:6, but further increasing the molar ratio to 1:7 caused a sharp decrease in conductivity (4.46 × 10^−1^ S/cm), although the performance increased by 92% [102]. CPs can undergo reversible redox reactions to switch between conductive and insulating states. This property is important for biosensors because it allows the detection of analytes on the basis of changes in electrical conductivity. Drawing on the chemical oxide polymerization mechanism, advanced manufacturing techniques like electrospinning, coating, and 3D printing have been explored for the production of electrodes from CPs [103]. While these methods above involving melting and thermoforming, and photocatalysis provide the benefit of increased productivity, they may introduce numerous additives into the final product, posing a noteworthy concern for manufacturing, especially probes from CPs [99,104].

#### 2.2.2. Electrochemical Polymerization

Electrochemical polymerization involves the introduction of three electrodes—a reference electrode, a counter electrode, and a working electrode—into a solution containing either reactants or monomers. Applying a voltage to these electrodes triggers a redox reaction that results in the formation of the polymer. Electrochemical polymerization can be further categorized into the following methods. The electrochemical synthesis of CPs can be carried out both at a controlled current and at a controlled potential of the working electrode. Also, using the method of cyclic voltammetry, CPs can be synthesized already at a potential (voltage) that changes during cycling. The primary advantage of electro(co)polymerization is the production of high-purity products. However, it is noteworthy that this method can produce only a small number of products simultaneously. Electrochemical polymerization is a cost-effective and environmentally friendly method for transforming monomers into CPs. The initiation of monomer polymerization is facilitated by the application of an oxidizing agent to the working electrode, which can be made of materials such as carbon [33], metal [105], and conductive glass. Notably, this polymerization method does not involve toxic chemicals, and it not only yields high-purity CP materials with outstanding electrical and electronic properties, but has a high level of reproducibility. Electrochemical polymerization comprises three steps. The first is the dissolution of the oligomer in the diffusion solution following the oxidation of the monomer, and the second is the deposition nucleation process. Finally, growth and chain propagation occur through polymerization. With the exception of the initial oxidation stage, the synthesis mechanism of each polymer is well understood because the process is governed by the synthesis parameters [106]. In the case of electrochemical polymerization, an understanding of the nucleation process and growth kinetics can facilitate the customization of polymer characteristics, including crystallinity, structure, morphology, and density, according to requirements. CPs feature a carboxyl or amino group that serves as an immobilization matrix for the covalent attachment of recognition molecules [107].

The polymerization process can occur through two potential pathways: the interaction between neighboring radical cations or the reaction between a neutral monomer and a radical cation. Various methods that induce charges from interactions are employed for the electrochemical polymerization of monomers and their derivatives. Several factors, such as the type of monomer, dopant, pH environment, electrolyte type, applied potential and potential window, initial scanning mode, solvent effects, and temperature, regulate the conductive properties of the resulting polymer film. The conductive properties, in turn, determine the applications for which the produced polymer film can be employed. The electrochemical polymerization method offers a unique advantage: the seamless integration of fabrication and modification in a single process. This versatile technique allows for precise control over the thickness of the polymer film and facilitates the production of monolayer or multilayer structures. Films with conductive or insulating properties can be prepared depending on the desired application. Notably, electroactive and conductive films produced using this method can be directly characterized without the need for additional purification. Furthermore, these CPs maintain their mechanical integrity while exhibiting metallic and semiconductor properties [108]. A significant advantage of this technique lies in the precise regulation of the initiation and termination processes by adjusting the oxidation potential applied during polymerization [109]. Additionally, the control of the electrode potential facilitates processes such as doping, rapid fabrication, the maintenance of film thickness, and deposition control, apart from helping achieve high cleanliness.

#### 2.2.3. Vapor-Phase Polymerization

In this method, polymerization occurs when the vapor-phase monomer is deposited onto an oxidant-coated substrate [110]. Polymerization reactions can be either chemical or electrochemical in nature, but they occur when the monomer is in the vapor phase. This method has several advantages over other techniques: it eliminates the need for solvents, minimizes the risk of agglomeration because the monomer is in vapor form, and enables the production of CPs with high electrical conductivity. Because of the homogeneity of the monomer, it can easily combine with volatile substances and coat uneven surfaces with a porous structure and interwoven fibers [111].

Vapor-phase polymerization has been used to synthesize PPy-coated filter paper for ammonia-sensing applications. Doping enhanced the conductivity of the filter paper from 1.78 × 10^−5^ to 3.34 × 10^−5^ S/cm. Consequently, the limit of detection (LOD) of ammonia vapor was 13 parts per million (ppm). Vapor-phase doping with HCl increased dopant localization along the PPy structure to create more active sites, which improved the LOD to 5.2 ppm [112]. A bacterial nanocellulose/PEDOT material was also prepared through vapor-phase polymerization. Environmentally friendly bacterial nanocellulose was used as a flexible substrate, and the highly conductive PEDOT polymer was introduced on the substrate to obtain composites with a sheet resistance 10 times lower (18 Ω/square) than those produced through polymerization in solution (188 Ω/square). The resulting material could be improved 100 times and rolled completely without significant loss of electronic performance. Furthermore, a bent bacterial nanocellulose/PEDOT film could be used as a green-light-emitting diode., indicating the high applicability of the conductive bacterial nanocellulose/PEDOT film [113]. The vapor-phase technique has been used to fabricate PANi nanotubes with high conductivity by combining aniline monomer and oxidant Mn_3_O_4_ nanofibers [114]. Furthermore, a PEDOT/graphene composite-based polymer film with high mechanical strength and electrical conductivity (310 ± 20 S/cm) was synthesized via vapor-phase polymerization on a Ta_2_O_5_ porous dielectric surface using FeTos/graphene [115]. Thiophene polymerization can be achieved through the vapor-phase polymerization of p-toluene sulfonic acid and other sulfonic acids that act as oxidants to polymerize PTh. The high electrical conductivities of the two types of PTh were observed to be 2.91 × 10^4^ and 3.75 × 10^4^ S/cm, respectively [116]. Table 1 lists the advantages and disadvantages of the main polymerization methods for CP synthesis, as well as the morphologies of the synthesized polymers [117,118,119].

**Figure 4 ijms-25-01564-f004:**
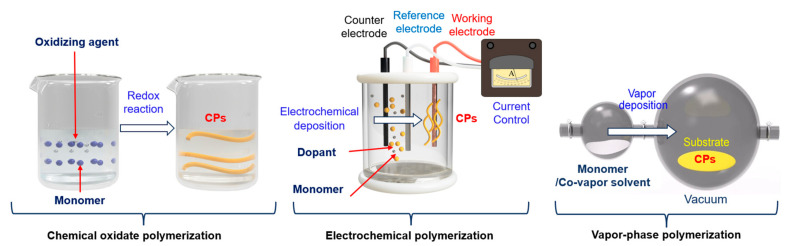
CPs polymerization methods. (Reprint from Ref. [118] Copyright (2015), with permission from Springer Nature Publisher).

## 3. Role and Advantages of CPs in Enhancing Sensor Performance

For tracking human health, the main function of a sensor is to effectively identify or respond to analytes or external stimuli such as chemicals, force, pH, temperature, humidity, and voltage selectively, with good sensitivity and linear response. Subsequently, the sensor should connect and transmit signals to other processing units. Most sensors used for monitoring the human body are passive and incorporate an analog signal receiver. An effective sensor design for human applications should feature a probe with a high sensitivity, good selectivity, fast response time, an appropriate operating range, reproducibility, and compatibility. In general, biosensors comprise three elements: a receptor biological recognition element, which is highly specific toward biological material analytes, integrated with or connected to a physicochemical transducer; a transducer part for converting the signal received from the biological target into an electrical signal; and an amplification and detection part for generating a discrete or continuous digital electronic signal that is proportional to the amount of a specific analyte or a combination of similar analytes [120]. Consequently, the transducer is the most crucial component that determines the effectiveness of the sensor because it directly determines the nature of the stimulating event with the stimulus and determines the operating principle of the sensor. In this component, CPs play a significant role, leveraging their unique electrical and chemical properties to convert recognition stimulation signals into electrical signals. Figure 5 shows some notable properties of CPs in comparison with other materials, highlighting their ability to achieve tunable conductivity across various applications. Most polymers fall within the semiconducting region (10^−7^ to 10^2^), which is inherently advantageous for electrochemical surfaces [87].

With advances in manufacturing techniques, the development of CPs for biosensors is based on fundamental principles surrounding polymer chain structure, morphology, combination ability, and signal reception mechanism signals and electrical conductivity. The π-conjugated system of CPs with electrons is localized along the polymer backbone. This delocalization allows them to conduct electricity when doped by chemical or electrochemical methods. By doping or de-doping, the conductivity of CPs can be controlled [96]. For applications in human health tracking, additional requirements are necessary for CP development. Currently, one research approach focuses on the synthesis of complicated CP structures such as poly (3, 3″dialkylquaterthiophene) (PQT12), poly(9,9-dioctylfluorene-alt-bithiophene) (F8T2), poly[2,5-bis(3-tetradecylthiophen-2-yl)thieno[3,2-b]thiophene] (PBTTT), poly[N-9′-heptadecanyl-2,7-carbazole-alt-5,5-(4′,7′-di-2-thienyl-2′,1′,3′-benzothiadiazole)] (PCDTBT), poly(phenylacetylene) (PPA), and poly(diketopyrrolopyrrole-co-bithiophene) (DPP4T) [125]. Another approach involves modifying suitable common CPs such as PAc, PPy, PANi, PPV, PTh, PEDOT, P3HT, and their various derivatives because of their excellent biocompatibility, stability, cost-effectiveness, and exceptional electrical and electrochemical properties, making them suitable for sensor design applications [14,15]. Table 2 shows the chemical and mechanical properties of CPs used in the field of sensor manufacturing, showcasing characteristic quantities such as conductivity and Young’s modulus.

While CPs are generally regarded as easily processable materials for biosensor fabrication, achieving optimal results requires continuous research to optimize their usage conditions. Therefore, designing biosensors using CPs requires careful consideration of several key properties to ensure effectiveness and reliability, such as selecting a CP with appropriate conductivity for a specific application. In this review, material fabrication and application to devices are influenced by many factors, especially in fields requiring high sensitivity, such as health-monitoring sensors. Any proactive change in these processes may introduce desired or unforeseen deviations in the characteristics of the CPs and may affect their applicability [126]. Therefore, the next section discusses important influences such as those of the CP characteristics, combination composition, and polymerization method on sensor function. It also explores the effect of the structure and morphology of the CP on the performance of the sensor. Furthermore, an assessment of the effect of the CP dimensions and size on the performance of the sensor is provided. This detailed review of CPs, their properties, and influencing factors aims to assist readers in identifying crucial components for producing highly efficient CPs for application in industrial devices and sensors. Composites derived from CPs or their combinations with other polymers exhibit enhanced mechanical properties suitable for specific applications. Moreover, CPs such as PPy and PANi have been investigated for use as conductive additives, particularly in conjunction with natural polymers, to address the challenges associated with the processing of CPs and to enhance the conductivity of insulating polymers [127]. On the basis of their function and the degree of interaction with the body, sensors used for human health monitoring can be divided into two main categories: biosensors and wearable sensors [128]. While most biosensors are quantitative or semiquantitative and involve biological or chemical sensing methods, wearable sensors operate on the basis of electromagnetic signals generated by the physiological activities of the body. Figure 6 describes the main properties and roles of CPs in human tracking sensors.

**Table 2 ijms-25-01564-t002:** Characteristics of CPs and their role in human health sensors.

Role of CP/Analyte Detection	Polymerization Method	Conductivity	Young’s Modulus	CP Advantages	CP Disadvantages
Transducer PPA/interleukin 6 sensing [129]	Vapor-phased deposition	10 to 104 S/cm	1.2 to 2.0 GPa	Easy synthesis, controllable, high electrical conductivity	Poor stability, non-thermoplastic
Transducer PPy/dopamine detection [130]	Chemical oxidative	40 to 200 S/cm [131]	430 to 800 MPa [131]	Biocompatibility, promotes cell proliferation, high conductivity, environmental friendliness, low cost, compatibility	Rigid, brittle, non-thermoplastic, nondegradable, insoluble in acetone, methanol, and ethanol solvents
Transducer PANi/pH detection [132]	Chemical oxidative	5 S/cm [133]	2.0 to 4.0 GPa [133]	Reversible doping, properties controllable by pH, wide range of conductivity, controllable conductivity, colored and transparent, various synthesis ways, stability with environment	Low processing capacity, inflexibility, lack of biodegradability, poor solubility, complex process
Transducer PTh/parathyroid hormone detection [134]	Vapor-phased deposition	10 to 100 S/cm [135]	3.0 GPa [135]	Low cost, high electrical, mechanical, and optical properties, high thermal and environmental constancy, smaller bandgap energy (2.0 eV) compared with PANi and PPy	Poor solubility with ordinary solvents, difficult to synthesize, poor chemical stability and processibility, poor flexibility
Transducer P3HT/immunoglobulin G detection [136]	Chemical, 3D printing	10^−2^, 10^−5^ to 10 S/cm [137]	28 MPa [138]	Low cost, high electrical, mechanical, and optical properties, high thermal and environmental constancy	Difficult process, poor solubility with common solvents, difficult to synthesize
Transducer PEDOT/temperature sensing [139]	Chemical oxidative	1200 S/cm [140]	1.2 to 4.0 GPa [140]	Transparent conductor, low redox potential, moderate bandgap, environmentally and electrochemically stable	Limited solubility, acidic nature, anisotropic charge injection, hygroscopicity

### 3.1. The Use of PAc in the Fabrication of the Transducer

#### 3.1.1. Effect of PAc Polymerization on Transducer Fabrication

PAc is a conducting polymer that is formed through the polymerization of acetylene monomers. The synthesis of polyacetylene involves chemical processes, typically initiated by doping with a suitable dopant or using a catalyst. Pac’s pure form is susceptible to oxidation when exposed to air and is also affected by humidity, which should be prevented in its application in wearable sensors [144]. However, each hydrogen unit on a polyene unit can be substituted with one or two replacements, resulting in a monosubstituted or disubstituted polyene unit to improve Pac’s stability. PAc can also be manufactured through the controlled combustion of acetylene in the presence of air to support a combination with other materials to enhance its properties. Electrodes incorporating PAc exhibit exceptional biocompatibility, high electrical conductivity, and a substantial specific surface area. Hence, PAc proves valuable in fabricating electrochemical sensors with remarkable sensitivity [145,146]. Both pristine PAc and its doped variations have been used for the detection of various substances, including glucose oxidase, colchicine, monoamine neurotransmitters, and their metabolites.

#### 3.1.2. Effect of PAc Structure and Morphology on Transducer Fabrication

The doped PAc structure is unstable, and its conductivity gradually decreases over time, presenting a major limitation of PAc. Some derivatives of PAc with a conjugated chain in a longer unit, such as PDA, PIP, or a derivative of PAc with a conjugated ring added to the structure (such as PPA), have a more stable structure [129,147]. However, PAc has many useful properties owing to its structure, including remarkable electrical conductivity, photoconductivity, a gas-permeable nature, the ability to form supramolecular assemblies, a chiral recognition capability to transform into helical graphitic nanofibers, and its behavior as a liquid crystal. As the first developed CP, it has undergone extensive studies on its conductivity changes, which exhibit a highly customizable range. It is commonly employed for preparing carbon paste electrodes [148] and constructing biosensors [21]. In addition, two disubstituted polyacetylenes with different side group of triazole and 2-(pyridin-2-yl)-1H-benzo[d]imidazole were built as sensor platform for glyphosate, an ingredient in pesticides. The structure shows a high degree of refinement of PAc by turning off the strong emission of PAc through Cu^2+^ and turning it on again in the presence of glyphosate at concentrations as low as 8 × 10^−8^ M. This result demonstrates the high customization and diversity of PAc’s architecture and opens a new approach for building sensing platforms based on PAc derivatives [149]. In addition, based on the change in shape of poly (phenylacetylene) with two aldehyde pendants to identify aliphatic amines and aromatic amines through visual color indicators, this research still has great potential without considering other factors. Physicochemical properties change with structure and are an open direction to develop sensing platforms for heterocyclic compounds such as amines [150].

### 3.2. The Use of PPy in the Fabrication of the Transducer

#### 3.2.1. Effect of PPy Polymerization on Transducer Fabrication

Among CPs, PPy and its derivatives have been extensively used in biosensor fabrication [151]. The development, synthesis, and polymerization of PPy have been achieved through the repeated oxidation of pyrrole by ferric chloride in methanol. Flexible conducting PPy films can be fabricated through various methods, such as electrochemical or oxidative methods using iron(III) trichloride (FeCl_3_). One of the advantages of the electrochemical route is the possibility of doping the polymer chains during the process, for example, with (Fe(CN)_6_)^3−^) [120]. Another attractive method is freezing interfacial polymerization, which can be used to obtain PPy films with a high electrical conductivity of 20 S/cm. The high conductivity results from the increased ordering of the CP structure [152]. Polymerization involves film formation on the anode. PPy is biocompatible, protects electrodes from fouling, and minimally disrupts the working environment. In some cases, PPy forms perm-selective films to exclude endogenous electrochemically active interferents. PPy exhibits numerous outstanding qualities, including excellent biocompatibility both in vitro and in vivo [107]. It exhibits robust chemical stability in both water and air [153,154], and features remarkably high electrical conductivity under physiological conditions [155,156].

The synthesis of PPy is straightforward and versatile, and it can be performed at room temperature in various solvents, including water. Furthermore, this material can be engineered to have a substantial surface area and adjustable porosity, which makes it highly adaptable for a wide range of internal and external sensing applications involving active biological molecules [157,158]. PPy exhibits responsiveness to external stimuli, allowing its properties to be controlled through the application of a voltage [159]. However, it is important to note that PPy presents certain challenges. Its molecular composition makes it intricate to manipulate because it tends to be resistant to high temperatures, mechanically rigid, and does not dissolve in solvents [160]. Various materials have been incorporated into PPy to enhance its flexibility, and its biocompatible properties have been leveraged to construct cell-based sensor systems that emulate the innate mechanisms of the body. These materials include muscle cells, fibronectin, titanium, bovine leukemia virus, protein, poly(ethylene terephthalate) (PET), tosylate, alginate ion, biotin, silk fibroin, and heparin conjugated with chlorpromazine [161,162,163,164]. Haung et al. conducted a notable investigation on the growth of PC-12 cells on PPy, which yielded intriguing results. PPy sheets subjected to electrical stimulation exhibited a cell growth rate approximately twice that of PPy sheets without electrical stimulation. This discovery underscores the potential of PPy in constructing neural conduits and suggests the possible use of electrical stimulation for enhancing nerve signal reception [165].

#### 3.2.2. Effect of PPy Structure and Morphology on Transducer Fabrication

The PPy structure with only one aromatic ring in the monomer unit of PPy exhibits high post-synthetic stability. Similar to PAc, functional groups with long carbon chains are added to the polymerization unit to enhance flexibility and free space in the overall polymer structure [166]. The synthesis of PPy involves blending it with various polymers. Studies have demonstrated that polylactide and poly(vinyl alcohol) (PVA) can be combined with PPy to obtain nanosized PPy composites. However, the PPy/PVA nanocomposites may exhibit reduced strength when exposed to water. To address this issue, certain solvents can be employed to uniformly disperse PPy in polylactide solution, resulting in uniformity and a low permeation threshold [167]. PPy nanocomposites have garnered attention because of their exceptional physicochemical properties related to nanoscale effects, and they show high stability and dispersibility even in highly viscous polymers [168,169]. Significant enhancements in PPy processability and functionality were achieved through copolymerization with other monomers possessing self-stabilizing functional groups. One research group discovered that electrochemical synthesis led to the formation of nanosized PPy sheets on electrodes. Although the size and morphology of these sheets can be controlled, the electrochemical polymerization process is not suitable for large-scale production. Therefore, an efficient method to synthesize nanostructured PPy without external surfactants or stabilizers is required.

One way to compensate for the shortcomings of a CP is to combine it with another polymer, thereby creating a composite with the positive qualities of both materials. PPy, known for its brittleness, has been deposited onto PET fabrics [170] and polyester [171] to enhance its flexibility. PPy-coated polyester fabrics demonstrate cytocompatibility and support cell growth after an initial period of low adhesion [172]. However, the conventional deposition method does not adhere well to PPy, resulting in the eventual release of the coating from the surface. This issue can be addressed using an alternative approach that involves covalently merging polyester fabrics with N-modified PPy, which exhibits significantly high resistance to delamination [171]. Combining poly(D,L-lactide) (PDLLA) and PPy, either deposited onto the surface of PDLLA as a film or incorporated into the PDLLA matrix as nanoparticles, yields a flexible, biocompatible, and biodegradable composite with improved conductivity compared with PPy-coated polyester fabrics. PPy/PDLLA maintains its electroactivity for up to 1000 h, and it supports the growth of fibroblasts [173,174]. Jang et al. produced PPy nanoparticles with sizes ranging from 2 to 8 nm through microemulsion polymerization, which was achieved by regulating the surfactant quantity [175]. In addition to surfactants, water-soluble polymers are often used to synthesize CP nanoparticles. The diameter of PPy nanoparticles prepared by this method is generally approximately 50–100 nm.

In addition, Li et al. developed a composite matrix of IL-PPy-Au. Three ionic liquid aqueous solutions based on imidazole with varying alkyl chain lengths ([Cnmim]Br, n = 2, 6, 12) were employed to synthesize the liquid ionic-polypyrrole (IL-PPy) material. The ionic liquid (IL) acted as a solvent during synthesis, facilitating the polymerization of pyrrole and the corresponding PPy derivative. Subsequently, Au microparticles were electrodeposited to enhance electrical conductivity and facilitate the immobilization of numerous biological molecules on the PPy electrode. The findings indicate that the introduction of ILs with longer chains not only significantly reduces the particle size of IL-PPy composites but also enhances the contact area, effectively increasing the number of folded conductive films on the electrode. This, in turn, expands the charge transfer area on the electrode and substantially improves the biosensor’s conductivity. The synergy of [C12mim]Br, PPy, and Au microparticles establishes an optimized platform for enzyme immobilization, proving to be an effective approach for fabricating highly sensitive biosensors. This work not only presents an advanced biomolecular immobilization matrix for biosensor fabrication with heightened sensitivity but also highlights the potential application of ionic liquids in CP synthesis [176]. In addition, research on the manufacture of PPy nanotubes and their use as electrodes has been conducted, achieving a maximum conductivity of 30.4 S/cm for a diameter of 95 nm [177,178].

### 3.3. The Use of PANi in the Fabrication of the Transducer

#### 3.3.1. Effect of PANi Polymerization on Transducer Fabrication

PANi is the second most extensively investigated CP [160,179]. Several chemical oxidative methods, such as enzymatic synthesis, polymerization under photo-stimulation, free-radical polymerization [180,181], and electrochemical methods [38], are commonly used for PANi synthesis. PANi exhibits redox activity, nonlinear optical characteristics, and high electrical conductivity coupled with proton and ionic transfer [38]. It also demonstrates stability in harsh chemical surroundings and a high thermal resistance. Depending on its degree of oxidation, PANi exists in different forms: fully oxidized pernigraniline base, half-oxidized emerald base, and fully reduced base. Among these, PANi is stable and conductive [160,179]. PANi offers many advantages: a simple synthesis method, good environmental stability, low cost, and the capability to electrically switch between conductive and resistive states [182,183]. Unfortunately, its use in sensing applications is limited owing to its low processability, lack of flexibility, non-biodegradability, and chronic inflammation when implanted in humans [182,184]. Polymeric materials, such as polyphenanthroline, have the unique capability of facilitating direct electron transfer between the active centers of biomolecules and electrodes [185]. In certain instances, a combination of two polymers, PPy and PANi, has been employed to immobilize enzymes on the electrode surface [120]. In the literature, various considerations are frequently discussed when selecting electrochemical biosensors and immunosensors, including factors such as simplicity, high sensitivity, robustness, mass production, miniaturization, multiplexing, and portability. This review excludes the application of CPs in the absence of biological recognition molecules to construct chemical sensors, except for nonenzymatic glucose detection scenarios, regardless of whether the target analyte is a biomolecule.

#### 3.3.2. Effect of PANi Structure and Morphology on Transducer Fabrication

PANi has gained widespread popularity as a CP owing to its straightforward synthesis and intriguing attributes such as chemical stability, flexibility, and ease of processing in solution. It is responsible for both p-doping and n-doping. The primary charge carrier in PANi can be manipulated by adjusting the pH of the dopant or by attaching organic or inorganic components to the polymer chain [101]. Furthermore, PANi can be synthesized with a wide variety of morphologies and electrical conductivities using both chemical and electrochemical techniques. The electrical conductivity of PANi is intimately linked to its oxidation state because PANi exhibits three distinct oxidation states corresponding to acid/base doping. Consequently, its electrical conductivity can be finely tailored within the range of 10^−7^ to 300 S/cm [186,187]. Its ease of preparation, high electrical conductivity, and high environmental stability make PANi suitable for use in sensor applications. For example, it can be used as a pH-switching electrically conducting biomaterial, an electrically active redox biopolymer, and a matrix for nanocomposite CP preparation. Methods have been developed for the preparation of PANi-based nanocomposite biopolymers. The electrical properties of PANi can be regulated by protonation or charge transfer doping.

Liu et al. developed a chemo-resistive sensor using PANi nanofibers adorned with Au nanoparticles for detecting volatile sulfur compounds in human breath. The Au/PANi gas sensor electrodes showed exceptional sensitivity to H_2_S and CH_3_SH [188]. The sensor successfully detected sulfur compounds in human breath after the consumption of raw garlic. Another study of the PANi nanofibers were horizontally ordered within the insulating gap region of an interdigitated electrode using a template-free electrochemical polymerization process [189].The Top of Form incorporation of Au nanoparticles onto PANi nanofibers was achieved through a redox reaction involving chloroauric acid and PANi in its emeraldine form. The resulting Au/PANi gas sensor electrodes exhibited exceptional sensitivity when exposed to hydrogen sulfide (H_2_S) at concentrations below 1 ppm and methyl mercaptan (CH_3_SH) at concentrations less than 1.5 ppm. The sensor’s capability to detect volatile sulfur compounds in human breath was demonstrated by exposing it to the exhaled breath of a healthy volunteer who had consumed raw garlic [190]. Han et al. synthesized zero-dimensional conductive nanoparticles using a chemical oxidative polymerization method. PANi particles, 20–30 nm in size, displayed remarkable conductivities of up to 24 S/cm. The synthesis involved octyl trimethylamine bromide and ferric chloride as the model compound and oxidizing agent, respectively [191,192,193].

### 3.4. The Use of PTh and P3HT Derivatives in the Fabrication of the Transducer

#### 3.4.1. Effect of the Polymerization of PTh and P3HT Derivatives on Transducer Fabrication

PTh is notable for its stable and high conductivity (10^3^ S/cm), which depends on the type of dopant and the polymerization process. PTh is inherently nontransparent and highly solvent-resistant [194]. Previous studies have investigated the impact of the conjugated sequence length in PTh on its conductivity. Specifically, oligomers composed of 11 thiophene units demonstrated conductivity comparable to that of higher-molecular-weight PTh. This aligns with the observation that short thiophene oligomers possess polymer-like characteristics, with the conductivity and carrier mobility increasing with the conjugation length up to the hexamer of thiophene. Transparency is a crucial attribute in applications that prioritize electrical conductivity, such as photographic films coated with antistatic materials, where a transparency level exceeding 90% is necessary [195]. The transparency of CPs has been enhanced through methods such as dilution, which can affect conductivity. Approaches used for diluting PTh include block copolymerization, grafting alkyl side chains onto the π-conjugated backbone, blending with a transparent polymer, and producing composites through the polymerization of thiophene absorbed in an insulating polymer.

Furthermore, methods such as plasma polymerization, electrochemical procedures, and the thin-layer deposition of PTh can be employed [196]. Plasma polymerization has the advantage of producing exceptionally thin defect-free layers that firmly adhere to various substrates without the need for solvents. Given that thiophene is an electron-rich aromatic ring that can be oxidized, highly conductive PTh can be obtained through p-doping. The electrochemical oxidation process promotes the formation of robust, adherent polymeric films, and the thickness of the polymer films can be varied by varying the polymerization duration in the electropolymerization process [197]. PTh has the advantage of being optically controllable, apart from exhibiting typical CP characteristics. In addition, the monomer can be easily functionalized, which facilitates the adjustment of optical properties along with conducting properties. PTh derivatives exhibit good processability in solution, which promotes uniformity in the fabrication process of PTh thin-film transducers [198]. In particular, because of the processability and solubility of PTh derivatives, their design, molecular weight, and π overlap between chains can be easily fine-tuned. The structure and degree of improvement in conductivity and processability in a solution depend significantly on the synthetic design and reaction pathways. The popularity of PTh serves as a leading example used to understand the importance of material synthesis [199]. A significant breakthrough has been achieved in the modulation of the formation of beta-substituted thiophene monomers and their sequential polymerization, resulting in solution-processable PTh derivatives [200]. Interesting results have been obtained in terms of improved conductivity after p-doping, and these results serve as the basis for exploring diverse potential applications for PTh-based materials.

The enhanced performance of PTh derivatives was achieved through the in situ polymerization of conjugated poly[(thiophene-2,5-diyl)-co-(benzylidene)]. This approach generates additional valence bonds on graphite and graphene oxide sheets, offering insight into the collective impact of the structure and doping on cyclic stability [33]. The intriguing and unconventional behavior of the less ordered, quasi-amorphous, conjugated polymers indicates their potential for attaining superior charge transport properties through electrochemical polymerization. In addition, a notable study focused on creating a polythiophene lactosylate biointerface and its interaction with Erythrina Cristagalli lectin, explored through differential pulse voltammetry. The polymerization of the 3-(3-azidopropoxy) thiophene monomer in the ionic liquid [Bmim][BF4] allows the postpolymerization structure to facilitate lactose or ferrocene grafting through a Cu(I)-catalyzed click reaction. This newly glycosylated CP biointerface serves as a foundation for developing a label-free, real-time electrochemical biosensor to investigate protein–carbohydrate interactions, enabling rapid protein analysis [201].

#### 3.4.2. Effect of the Morphology and Structure of the Polymerization of PTh and P3HT Derivatives on Transducer Fabrication

The molecular structure and fine-tuning of the crystallinity in PTh are crucial for achieving high charge mobility in the transducer. High electrical conductivity in PTh can be realized by forming conductive PTh layers on a dielectric substrate [202]. PTh can function as a layer that is suitable for metal electroplating, which has been difficult to achieve with insulating substrates [33]. In particular, P3HT can undergo recrystallization in the presence of common non-crystalline polymers such as PS and PMMA to form nanofibrous composite films. Shimomura et al. produced transparent conductive P3HT nanofibers using polymer composites with PMMA and AuCl_3_ oxidation solution [203]. Furthermore, flexible conductive membranes with high electrical conductivity have been realized by appropriately adjusting the optimal ratio of acetonitrile and a boronic agent [204]. Notably, PTh serves as an activating layer that facilitates the formation of smooth, conductive nickel nanoparticles on an insulating matrix [205]. The stability of PTh-functionalized multiwalled carbon nanotube (MWCNTs) binary composites has been achieved using sodium bis(2-ethylhexyl) sulfosuccinate micelles prepared via oxidative polymerization [206]. The combination of binary mixtures and the incorporation of entangled silver nanoparticles has paved the way for the development of ternary nanocomposites. These ternary composites exhibit superior performance, with an enhanced electrical conductivity of 80.76 S/cm. This improvement can be attributed to the efficient charge transport facilitated by the PTh interlayer, which effectively serves as a conductive bridge between MWCNTs and silver nanoparticles [33]. Enhanced conductivity plays a pivotal role in promoting accurate sensing, because analytes respond to variations in conductivity or resistance [36].

In addition to improvements in conductivity, sensitivity, and processability, mechanical flexibility is an equally vital consideration for the application of PTh in transducers. Achieving desirable mechanical properties has proven to be a significant challenge for PTh derivatives, such as P3HT [207], an alkylated derivative of PTh, and widely investigated tetrafluoro-7,7,8,8-tetracyanoquinodimethane p-conjugated electrically of CP. P3HT exhibits commendable solubility in a variety of organic solvents, contributing to its superior film-processing properties. Furthermore, it is commercially available [208]. Because P3HT has been successfully used in all-organic solar cells and all-organic field effect transistors, research groups have investigated its potential as a thermoelectric material. Crispin et al. evaluated the thermoelectric properties of P3HT films doped with nitronium hexafluorophosphate [209]. Maximum performance was observed in a sample with a 31% doping level. It was also demonstrated that bulky PF6 anions inhibit the formation of crystallites at low doping levels, but with an increasing doping level, the structural order increases in line with the electrical conductivity. This challenge has impeded progress in organic electronics and their industrial implementation. Typically, semiconductive-based polymers exhibit limited stretchability because of their high crystallinity resulting from their rigid molecular structure and strong π-π interactions. This has led to a growing demand for CPs with improved intrinsic stretchability [210].

Both PTh and P3HT, while flexible and bendable to a certain extent, lack adequate robustness to repeated bending processes. This mechanical instability can introduce fluctuations in the electrical properties of P3HT-based flexible devices [141]. To address this issue, researchers have developed high-molecular-weight carbon chains with disiloxane moieties in the side chains to enhance the tensile strength. Moreover, investigations into the molecular structure and physical properties of substituted thiophene have confirmed the outstanding mechanical flexibility of certain derivatives. The impact of side chains on chain mobility and the glass transition temperature (T_g_), a measure of flexibility, in prepared substituted polymers has also been investigated [211]. Specifically, bulkier side-chain substituents tend to increase stiffness, resulting in a higher T_g_ because bond rotations are constrained. This restriction on bond rotation can be achieved by using branched side chains, such as the P3HT isomer poly(3-2-methylpentylthiophene), which features a methyl-branched side chain instead of the typical linear chain polymers. The inclusion of a methyl group in the side chain limits the rotation of adjacent C-C bonds, rendering the side chain stiffer than the linear chain in P3HT. Branched side chains restrict chain movement and reduce crystallization-induced phase separation, which is a crucial factor determining the stability of the transducer layer in sensors [212]. As shown in Figure 7, Shin et al. presented a straightforward approach for enhancing the NO_2_ sensing capability of an organic field-effect transistor (OFET) sensor operating at room temperature. The approach involved incorporating a nano P3HT film with reduced graphene oxide (rGO) through phase-supported cutting and a coating separation technique. The synergy between the nanoporous P3HT, which serves as a pathway for analyte diffusion, and the rGO, which acts as an adsorption site, leads to significant changes in the electrical properties of the nanoporous P3HT/rGO OFET when it comes in contact with NO_2_ gas molecules. This underscores the potential of OFETs as effective NO_2_ sensors. More specifically, novel nanoporous OFET sensors featuring rGO-integrated nanoporous P3HT membranes exhibit a significantly improved response, with rGO amounts of approximately 61.3%, when exposed to 10 ppm NO_2_ gas. In contrast, sensors based on nonporous P3HT/PS/rGO composite membranes showed a response of approximately 17.7%. Moreover, the novel nanoporous OFET sensors exhibited exceptional response and recovery characteristics, with a response time of approximately 62 s and a recovery time of approximately 145 s. They also showed high sensitivity at approximately 1.48 ppm^−1^ and excellent selectivity [213].

Another approach to enhance the mechanical properties of PTh and P3HT involves substituting side chains with esters, which provides greater freedom of movement. This substitution with ester side chains leads to extended two-way conjugation, thereby improving mobility and stretchability by strengthening the amorphous structure with reinforcing agents [214]. Notably, the orientation of the crystals during the drawing process plays a crucial role in determining the conductivity. PTh disiloxane-substituted derivatives exhibit a specific stable crystallographic orientation, even when strain degree approximately 140% [215]. Control over the crystallization modes of conjugated block copolymers based on poly(3-dodecylthiophene) (P3DDT) and poly(2-vinylpyridine) can be achieved by controlling through the regioselectivity of P3DDT, as well as the melting temperature and crystallization rate using P3DDT. Such control involves poly(2-vinylpyridine) at low recovery rates and crystallization at temperatures near or below T*_g_* [216]. Crystal growth is limited by the cylindrical block copolymer structure or the glass layer [216]. In a study focused on the formation of a stretchable active channel matrix, a mixed solution of PS-block-poly(ethylene-co-butylene)-block-polystyrene and P3HT was obtained through rotational molding. This process facilitates the in situ phase separation of P3HT nanofibers on the surface of the rubber matrix, the assembly of nanofibers into wide bundles, the formation of networks of these bundles, and the indentation of the bundles on the active rubber surface [217].

In various studies, researchers have explored different approaches to enhance the performance of conducting polymer-based devices. Chen et al. adopted a block copolymer strategy to create resilient memory devices, demonstrating consistent performance under diverse conditions [218]. In this context, Higashihara et al. utilized Kumada–Tama catalytic transfer polycondensation and living polymerization to synthesize a triblock copolymer, achieving morphologically adjustable and elastic polymer matrices [219]. Similarly, Watts et al. produced click-processed rod-coil diblock copolymers by combining alkynyl-functionalized P3HT with azido-terminated PBA homopolymers. Furthermore, they examined the effect of bulk ratios on mechanical and morphological characteristics, especially for CP application in stretchable field-effect transistor (FET) devices [220]. In a study comparing the electronic properties and surface morphologies of P3HT and 2,3,5,6-tetrafluoro-7,7,8,8-tetracyanoquinodimethane (F4TCNQ) films doped directly in solution or through sequential doping processes [221], Joeng et al. observed that the sequential doping technique significantly increases the electrical conductivity at the equivalent doping ratio [97]. Additionally, Jacobs et al. conducted a comparative study on the electronic properties and surface morphologies of P3HT films doped with F4TCNQ, highlighting the significant enhancement in electrical conductivity achieved through sequential doping. The optimized morphology, achieved by stacking alternating layers of the polymer and dopant, resulted in thicker films with properties closely resembling their thin-film counterparts, showcasing the potential for fine-tuning these processes to improve device performance [222].

In a study by Hynynen et al., as shown in Figure 8, the sequential doping approach was employed by exposing a P3HT film to F4TCNQ vapor. They demonstrated that the crystalline structure of the film could be modulated by varying the doping level [223]. Their work also revealed that the molecular weight of P3HT had a relatively modest effect on the film conductivity, and they reported an electrical conductivity of 12.7 S/cm [19]. A recent approach for doping P3HT films involves the use of organic molecular dopants such as electron acceptors. These dopants induce ground-state charge carrier transfer in the semiconductor polymer host, creating polarons and bipolaron after that [224]. However, dopant anions remain in the film and often result in drastically reduced solubility. Therefore, aggregation is a major issue in polymer dopant liquid mixtures. To avoid such phenomena, P3HT solutions should be diluted and maintained at a high temperature to prevent aggregation. Using conventional solution processing, a P3HT film achieved an electrical conductivity of 4 × 10^−4^ S/cm [225]. A problem with bulk doping methods is that large dopant amounts, such as rGO, adversely affect the morphology of the P3HT film and lead to aggregation, which inhibits the achievement of higher conductivity levels [226]. Figure 9 shows the performance of Hynynen’s OFET sensor.

### 3.5. The Use of PEDOT in the Fabrication of the Transducer

In contrast to many other CPs, PEDOT offers superior conductivity, higher transparency, and exceptional environmental stability. Nevertheless, this material has drawbacks: it is insoluble in both water and organic solvents. This makes it challenging to use in molding or spin coating techniques. To address the solubility and processability issues, Bayer developed a graft copolymer of PEDOT with PSS polymerized in an aqueous colloidal suspension. This innovation has provided an effective solution to the processing difficulties associated with PEDOT, making PEDOT:PSS suitable for use in sensor fabrication [140,228].

#### 3.5.1. Effect of PEDOT Polymerization on Transducer Fabrication

PEDOT is one of the most extensively researched CPs, primarily because of its exceptional stability and high electrical conductivity. The initial enhancement of this polymer focused on achieving a soluble CP without the presence of α,β- and β,β-coupling in its backbone. PEDOT can be synthesized using standard oxidative or electrochemical polymerization techniques [25,207]. It exhibits transparency within the matrix and exceptional stability in its oxidized state. The behavior of PEDOT, which encompasses aspects such as solubility and stability, has been investigated by incorporating a water-soluble polyelectrolyte. PSS served as a dopant and facilitated charge-balancing in the polymerization process to form PEDOT:PSS. The synergy between PEDOT and the PSS electrolyte resulted in several notable properties, such as water solubility, a high conductivity of approximately 10 S/cm, excellent architectural light transmittance, and remarkable stability [229]. High-density PEDOT:PSS can withstand prolonged exposure to a 100 °C air temperature for over 1000 h, with minimal changes in its electrical properties. It is noteworthy that the reported conditioning of PEDOT nanoparticles mainly involved the relative solubility regime of the 3,4-ethylenedioxythiophene (EDOT) monomer in aqueous solutions. PEDOT thin films can attain impressive conductivity levels, with values reaching up to 6259 S/cm, whereas single-crystal nanowires can surpass this level, reaching 8797 S/cm. PEDOT synthesis typically involves three primary polymerization methods: oxidative chemical polymerization involving EDOT-based monomers in the presence of various oxidants, electrochemical polymerization conducted in a three-electrode setup using EDOT-based monomers, and transition-metal-mediated coupling techniques. PEDOT itself is insoluble, and the addition of other polymers is often necessary to enhance its solubility and overall processability. PDMS is the most commonly used material for this purpose, and it enhances PEDOT’s stability in aqueous environments. Because of their remarkable biocompatibility, PEDOT and its derivatives, such as PEDOT:PSS, have found widespread applications in diverse biomedical fields, including bone, heart, and nervous tissue engineering, as well as drug delivery systems [230,231].

In biosensors, PEDOT:PSS, in combination with graphene oxide (GO), is used as a conductive substrate for the immobilization of glucose oxidase to perform enzyme-based glucose detection. More recently, PEDOT:PSS and GO have been used in gold microelectrodes, leading to improved electrochemical, biochemical, and mechanical properties of the microelectrodes and making them suitable for neural implant applications. PEDOT has been electrochemically deposited onto biodegradable magnesium microwires, and the microwires have been used for recording nerve signals. These PEDOT-coated Mg microwires were further spray-coated with poly(glycerol sebacate) for use as an insulating layer. The resulting microelectrode exhibited performance comparable to that of equivalent platinum (Pt) microelectrodes commonly used in clinical settings. The PEDOT-coated microelectrode exhibited superior electrical properties compared with the Pt electrode, with a charge storage capacity five times that of Pt and lower impedance within the frequency range of 1 MHz to 0.1 Hz. Furthermore, the Mg-based electrode demonstrated similar neural recordings in vivo [232].

#### 3.5.2. Effect of PEDOT Morphology and Structure on Transducer Fabrication

Fully organic implants have been developed using PEDOT, with ongoing research focused on improving their biocompatibility and biodegradability. For instance, the Ferlauto group fabricated an all-organic transient neural probe consisting of a polycaprolactone substrate and a packaging material, with PEDOT:PSS-ethylene glycol as the electrode material [233]. These electrodes were implanted in the visual cortex of mice, and neural activity was measured during rest, induced seizures, and visual stimulation. The study demonstrated the electrodes’ long-term effectiveness, and they remained functional months after implantation. While PEDOT:PSS is not inherently biodegradable, the authors hypothesized that electrode degradation is related to hydrolysis, which is ascribed to the electrode’s adsorption of hydrogen peroxide in the environment [234,235]. They also suggested that the transient probe resulted in a less pronounced glial scar compared with no transient polyimide probes, facilitating microglia access for the phagocytosis of delaminated PEDOT:PSS. Furthermore, the results indicated the complete degradation of the electrode after 1 year at 37 °C and pH 12, with an acceleration factor of approximately 2.5 compared with pH 7.4 [236]. This gradual degradation was also evident when electrode implantation resulted in only a small glial scar after nine months. In another study, Pradhan et al. developed a fully organic, biocompatible, and bioabsorbable temperature sensor using silk and PEDOT:PSS. Silk, a natural protein obtained primarily from silkworms, was utilized as the sensor’s substrate and housing material. The conductive layer was composed of PEDOT:PSS dispersed in photoreactive sericin, a biodegradable variant of silk. This dispersion in biodegradable sericin provided the controllable degradation of PEDOT:PSS, allowing the entire sensor to completely degrade within 10 days in a protease solution. This technique was also employed to fabricate PEDOT:PSS-based silk sensors for glucose, dopamine, and ascorbic acid detection [25]. Lupu et al. demonstrated sinusoidal voltages for the enhanced incorporation of enzymes into polymers during electrolysis, developing a tyrosinase-based dopamine biosensor attached to a PEDOT membrane [237]. Table 3 shows the effects of CP polymerization and morphology on sensor performance.

Constructing an effective biosensor system requires certain static and dynamic requirements. Adhering to these specifications allows the optimization of biosensor performance in commercial applications. The major parameters include selectivity, sensitivity, linearity, response time, reproducibility, and stability. Selectivity is a critical feature when choosing a bioreceptor for a biosensor. The bioreceptor should be capable of detecting a specific target analyte molecule in a sample containing a mixture of substances and undesired contaminants. Sensitivity is the ability to correctly detect and identify the smallest amount of analyte in a minimal number of steps, even at low concentrations (ng/mL or fg/mL). Linearity contributes to the precision of the measured results. The greater the linearity, represented by a straight line, the higher the detection capacity for various substrate concentrations. Response time is the duration required to obtain 95% of the results. Reproducibility indicates the biosensor’s capability to yield identical results in successive measurements of the same sample. The last parameter, stability, is a vital characteristic in biosensor applications involving continuous monitoring. It indicates the degree of susceptibility to environmental disturbances both within and outside the biosensing device. Factors influencing stability include bioreceptor affinity (the extent of analyte binding) and degradation over time [235,264].

## 4. Advances in the Use of CPs in Biosensors

In biosensor design, the immobilization element plays a major role in the performance characterization of biosensors. To achieve good sensitivity, longevity, and an extended operating period, biomolecules must directly attach to the biosensor surface, as shown in Figure 10. The most commonly used methods to immobilize biomolecules in polymers are physical adsorption, covalent binding, and trapping [120]. The choice of immobilization strategy mainly depends on the specific biological factor. For instance, antibodies and DNA [265] are preferably immobilized by adsorption or covalent binding on the surface of CPs membranes to facilitate analyte access to these biological recognition molecules. Conversely, the trapping method is commonly used to immobilize oxidoreductases in polymer membranes to facilitate electron transfer from the redox center of the enzyme to the analyte solution surrounding the CPs and enables the redox reaction of electroactive substances, such as hydrogen peroxide [19]. The CPs, characterized by exceptional electrical conductivity, high electronic affinity, and a low ionization potential, plays a pivotal role in enhancing biosensor performance. CPs can be readily polymerized on the electrode surface using noble salts or dopants to enhance conductivity and catalytic activity and increase the surface area. Achieving high conductivity is always desirable because it minimizes the interface resistance between electrolytes, resulting in a more robust signal response. In a broader context, CPs including π-conjugated polymers not only offer mechanical support to sensor membranes, but also significantly enhance sensitivity and stability in sensing devices while ensuring structural integrity. Furthermore, their excellent biocompatibility makes them ideal matrices for biosensors, and they facilitate the stable immobilization of biomolecules on their surfaces while preserving their activity. Regular immobilization methods include cross-linking, covalent binding, physical entrapment, and adsorption.

The covalent immobilization method uses functional groups found in biological molecules (-COOH, -NH_2_, -SH) to establish strong bonds with CPs. This involves, for instance, the formation of amide bonds between amide groups in biological molecules and carboxylic groups present on a CPs. Furthermore, the covalent attachment of biomolecules is often accomplished by initially synthesizing monomers that are functionalized with amino side groups, followed by polymerization of the functionalized monomers [266]. Functionalization introduces side functional groups to the surface of CPs films, allowing them to covalently bind to biomolecules containing corresponding functional groups containing sulfur and nitrogen. Frequently, the formation of covalent bonds between biomolecules and polymers is induced by using cross-linking agents such as 1-ethyl-3-(3-dimethylaminopropyl) carbodiimide or glutaraldehyde. The covalent immobilization method offers the advantage of low diffusion resistance, and it strongly binds polymers and biomolecules, thereby reducing biomolecule loss. Consequently, electrodes produced using this method tend to be more stable over time, although maintaining biomolecule activity can be challenging in certain cases [267]. Conversely, the adsorption mechanism is a simpler process that involves the physical adsorption of biomolecules onto the polymer surface. At times, the presence of opposite charges on the CPs and biomolecule assists in biomolecule immobilization. For instance, positively charged PANi-PIP membranes at pH 4.5 can adsorb negatively charged glucose oxidase, making them highly sensitive to changes in glucose concentration [268]. This method has the advantage of causing minimal disruption to the natural structure and functions of biological molecules, often leading to highly sensitive reactions. However, a significant drawback is that the direct adsorption of biomolecules onto the surface of CPs often results in low sensor stability, as biomolecules can leach from the surface in response to environmental changes (such as pH or ionic strength), although the use of polymer membranes can mitigate this leaching [269,270].

Another widely used method for immobilizing antibodies, enzymes, and DNA is entrapment. It involves the preparation of an electrolyte solution containing both monomers and biomolecules. Therefore, a polymer film containing biomolecules is formed at the electrode surface. Entrapment is an interesting technique because it strengthens the adhesion between the biomolecule and the polymer membrane in a single step. Furthermore, this strategy helps to easily control the number of entrapped biomolecules by regulating the thickness of the CPs layer. Entrapment generally enhances the sensitivity and lifetime of biosensors. However, entrapment can lead to issues related to the inaccessibility of the drive-in biomolecule [271,272,273,274]. Furthermore, for certain CPs, extremely acidic conditions or high oxidation potentials are required during their electrochemical preparation, which are generally unsuitable for accommodating biomolecules. During electrolysis, supporting electrolytes are frequently introduced to enhance the conductivity of the monomer solution. However, these electrolytes often vie with biomolecules for polymer doping sites, resulting in a diminished capacity for trapping biomolecules. This issue is particularly challenging when dealing with expensive bioentities. A potential resolution to this challenge involves employing biomolecules as counterions in the preparation of CPs membranes, a strategy previously demonstrated with PPy and GO enzymes, which facilitates more effective entrapment [275,276].

Biosensing is categorized based on two primary targets: biochemicals and aptamer agents. When CPs is used as a sensitive material in an electrochemical sensor, an attempt to identify the target analyte using a biosensing base embedded within a polymer conduit will produce a measurable signal. The cumulus analysis method can be used to convert the signal into an electrical signal. The introduction of a CPs provides notable advantages, such as enhancing the sensitivity and filtration level of the biosensor while minimizing the effects of interfering substances [8]. The filtration efficiency of a biosensor relies heavily on the interaction of the analyte with the receptor on the CP base. Ensuring the quality of the pupil in the CPs and the adhesion of the CPs to the surface of the biosensor are important for ensuring the long-term effectiveness of the biosensor. The sensitivity is influenced by many factors, with the intensity of the electrochemical signal generated during the interaction between the analyte, bioreceptor, and CPs being a key factor. This electrochemical signal can manifest as a change in the voltage, current, electrical/return path, impedance, or electron mass exchange oxidation or deodorization reactions. These variations can be used in special types of biosensors, such as voltametric, impedimetric, conductimetric, amperometric, and potentiometric biosensors [128]. In determining the effectiveness of a CPs biosensor, striking a balance between material properties, functionality, surface morphology, and response results is crucial for accurate and reliable detection. The key parameters for evaluating biosensor performance include sensitivity, stability, resolution, detection time, limit of detection, linear dynamic range, reproducibility, and dynamic range (see Table 4).

### 4.1. Biochemical Sensor

The presence of biological molecules constitutes the main mechanism in various body biochemistry activities, emphasizing the crucial importance of identifying these biological molecules. An FET sensing platform was fabricated to detect specific biological entities, and a buffer solution was used as an ion gate liquid based on the controlled functionality of carboxylic-acid-functionalized (CA)-PPy nanotubes (CPPy). The PPy nanotubes were covalently immobilized on the microelectrode platform to establish good electrical contact with the metal electrodes, and the aptamers were linked to the PPy nanotube surface through covalent bonds as a factor for molecular identification. The selective recognition ability of thrombin aptamers combined with the charge transport properties of CPPy nanotubes facilitated direct electrical detection, with an LOD value of less than 50 nM. Notably, no label was required for thrombin protein. Figure 11 shows the CPPy transducer design for the aptamer sensor.

Gliga et al. [146] pioneered the development of a sandwich-type sensor for the rapid and specific detection of adenosine. Their innovative configuration involved constructing a covalently imprinted biorecognition element using two superimposed polymer films. The foundation comprised a synthetic boronated layer alongside a CPs of poly(bithiophene) structure, with an additional layer containing poly(3-indolacetic acid) and featuring carboxyl groups. This combination of materials enhanced the sensor’s selectivity toward adenosine. The biomimetic sensor layer was meticulously fabricated through a two-step process, featuring rapid and precisely controlled electropolymerization. This process, which requires only a few minutes for each polymer membrane, depends on the multipoint anchoring of the sample. The outcome is a network of highly precise imprinted sites that can selectively and reversibly bind to adenosine through both spatially directed covalent and noncovalent interactions. This feature allows the use of multiple sensors for detecting biological transformations. Remarkably, this sensor demonstrates a wide dynamic range, spanning from 0.37 µM to 37.4 µM, and exhibits an impressively low detection limit of 0.21 µM. These capabilities have been successfully employed for the quantitative determination of adenosine in biological samples, specifically urine, with outstanding recovery rates [18]. Puttananjegowda et al. presented an electrospun nanofibrous membrane of silicon carbide nanoparticles with PEDOT:PSS for electrochemical enzymatic glucose sensing. The PEDOT:PSS fiber layer immobilized silicon carbide and enhanced glucose enzyme binding. The glucose concentration in a 5 mM potassium ferricyanide electrolyte was detected as +0.6 V. The glucose-based electrodes exhibited a detection range from 0.5 mM to 20 mM, with a sensitivity of 30.75 µA/mM cm^2^ and a detection limit of 0.56 µM, along with a durability of up to 50 days [292].

A straightforward manufacturing method involving the SAPS technique has been employed to prepare nanoporous P3HT composite membranes integrated with rGO following the SAPS technique. In the resulting composite membranes, rGO was uniformly dispersed and exposed in the P3HT pores, and a robust charge transfer interaction occurred between P3HT and rGO. Upon exposure to 10 ppm of NO_2_ gas, significant changes in charge mobility and current were observed in the nanoporous P3HT/rGO composite membrane, with values of approximately 212.3% and 114.3%, respectively. The observed values were notably higher than those of the bare nanoporous composite P3HT film (∼109.1% and ∼18.5%, respectively) and the P3HT/PS/rGO composite film (∼138.2% and ∼54.5%, respectively). This highlights the superior sensing performance of the nanoporous P3HT/rGO film. Notably, rGO exhibited better recognition capabilities for NO_2_ than P3HT, indicating that morphological characteristics are more critical than the charge transport capacity of the active layer in determining the sensing performance. In a comprehensive assessment of sensor performance, the nanoporous P3HT/rGO OFET demonstrated exceptional results: a response of approximately 61.3%, response/recovery time of approximately 62 s/145 s, and sensitivity of approximately 1.48 ppm^−1^ to NO_2_ gas. This performance surpassed that of the P3HT/nonporous PS/rGO OFET (∼17.7%, ∼102 s/147 s, and ∼0.62 ppm^−1^, respectively). The outlined approach holds significant promise for producing nanoporous CPs composite thin films suitable for high-performance OFET sensors in a variety of environmental and healthcare applications [213]. Figure 12 illustrates a straightforward and scalable technique proposed by Heo et al. to fabricate controlled structural features within polymers. This technique involves incorporating bis(2-ethylhexyl) sulfosuccinate (MAOT) along with Na as a counteragent into the polymer phase, resulting in the formation of an integrated pore structure in the polyacrylonitrile (PAN) nanofibers. The use of MAOT with Fe as the counteragent leads to the development of distinctive iron-in-hole structures in PAN nanofibers, referred to as FeCNF. These FeCNF structures exhibit superior performance in terms of charge storage and water separation. The MAOT-in-polymer patterning technique can be extended to include various metal precursors, and it can be applied for CP production for making porous fibrous electrodes [293].

### 4.2. Apta Sensor

The biomarker Mucin 1 (MUC1) plays a crucial role in the metastasis and invasion of various cancer types. Detecting MUC1 in fluids and performing in situ imaging of MUC1 at the cellular level are crucial for early cancer diagnosis and disease progression monitoring. To address this need, we developed a PDA liposome-based sensor system functionalized with a Cy3-labeled MUC1-binding aptamer (Cy3-Apt). This system was intended to serve as a “turn-on” fluorescence nanosensor for MUC1. In this sensing system, the fluorescence of Cy3-Apt is initially quenched through energy transfer between the fluorogenic conjugated backbone and PDA. However, in the presence of MUC1, the aptamer recognizes it, inducing a structural transition in both the PDA liposome and the aptamer, ultimately restoring the red fluorescence. This nanosensor is suitable for the sensitive and selective analysis of MUC1 in aqueous media, with a detection limit of approximately 0.8 nM. Furthermore, the effectiveness of this nanosensor was verified by mapping the spatial expression of MUC1 in cancer cells, and it was found to have excellent sensitivity [294]. Furthermore, a novel PEDOT/FeOOH/BiVO4 photoactive nanohybrid with excellent photoelectrochemical (PEC) performance was assembled to fabricate an ultrasensitive biosensor for microRNA-375-3p (miRNA-375-3p) detection [295,296].

The PEDOT/FeOOH/BiVO_4_ nanohybrids exhibited a markedly enhanced photocurrent compared with the traditional FeOOH/BiVO_4_ photoactive composite. This improvement can be attributed to the surface charge separation promoted by PEDOT, which was not only used to improve the substrate’s electrical conductivity but also acted as a local photothermal heater to enhance photogenerated carrier separation. Based on this PEDOT/FeOOH/BiVO_4_ photoelectrode and an enzyme-free signal amplification strategy involving a target-sensing catalytic hairpin assembly (CHA) and a hybrid chain reaction (HCR), a PEC sensor platform for the detection of miRNA-375-3p was developed, and it had a linear range from 1 fM to 10 pM, with an LOD of 0.3 fM. Furthermore, this review provides a general photocurrent enhancement strategy to develop high-performance PEC biosensors for the sensitive detection of biomarkers and early disease diagnosis. PEDOT/FeOOH/BiVO_4_ nanohybrids with excellent photovoltaic performance promoted by the photothermal effect were developed to construct a sensitive PEC sensor for detecting circulating microRNA cancer biomarkers. A PEDOT/FeOOH/BiVO_4_ photoanode was fabricated to achieve a high-intensity initial photocurrent. After the immobilization of PolyA-HP1 on the electrode by deposited Au nanoparticles, the target miRNA-375-3p hybridizes with PolyA-HP1 to generate a single-stranded fragment that reacts with HP2 to form a PolyA-HP1-HP2 DNA duplex. The released target could participate in opening the remaining PolyA-HP1 on the photoelectrode. PolyA-HP1-HP2 is formed with a special binding site that further hybridizes with HP3 and HP4 to produce long double-stranded (dsDNA) polymers (PolyA-HP1-HP2-(HP3-HP4)n). Therefore, a targeted amplification strategy involving CHA and hybridization chain reaction was used to convert a minimal amount of miRNA-375-3p into dsDNA. This immobilized methylene blue acted as an intercalator to promote electron transfer, resulting in a significantly enhanced photocurrent signal for the quantitative detection of miRNA-375-3p. The PEC biosensor exhibited remarkable sensitivity due to the high initial photocurrent of the PEDOT/FeOOH/BiVO_4_ photoanode and the target-induced amplification strategy. The signal amplification strategy designed by integrating target-induced CHA and HCR significantly improved the sensitivity of the PEC biosensor. Recoveries for 10 fM, 100 fM, 1 pM, and 5 pM targets ranged from 94.89% to 106.6%, and RSD values ranged from 5.72% to 9.51%, demonstrating the reliability and potential of the fabricated PEC biosensor in clinical biomedical analysis. The mechanism and process for making a biosensor based on PEDOT are shown in Figure 13 [297].

In another study, a biosensor was designed to resist fouling, with the capability of directly analyzing circulating tumor cells (CTCs) in blood. This biosensor employs an engineered multifunctional peptide in conjunction with a PEDOT transducer. The engineered peptide demonstrated antifouling properties in complex biological environments and specific recognition capabilities for capturing MCF-7 breast cancer cells. The electroplated PEDOT enhances electron transfer at the sensor interface, thereby improving the signal-to-noise ratio for detection and ultimately enhancing the biosensor’s sensitivity. The integration of the multifunctional peptide with PEDOT ensures that the biosensor can function directly in blood samples, eliminating the need for purification or separation processes. The antifouling electrochemical biosensor designed for detecting MCF-7 cells exhibits a broad linear range spanning over four orders of magnitude, with an LOD of 17 cells/mL. Notably, even when operating in the presence of 25% human blood, the biosensor maintains a linear response with an LOD of 22 cells/mL, showcasing resilience against biofouling in real blood scenarios. This design provides a promising strategy for the direct analysis of CTCs in human blood without the need for complex pretreatment. The developed biosensor has potential practical applications in the field of liquid biopsies for cancer, offering a straightforward and effective method for cancer detection [298].

## 5. Use of CPs in Wearable Sensor Design

CPs and their composites have various wearable device applications because of their flexibility and electrical conductivity. In contrast to biological sensors, wearable devices use CPs as the main material for shaping and recording signals through the inherent mechanisms of the CPs. Current research directions predominantly involve combining CPs with other materials to enhance or supplement properties such as biological, mechanical, and optical compatibility, with the aim of increasing the performance of CP-based sensors [296]. Although some breakthroughs have been achieved with carbon-based nanostructures [299], including carbon quantum dots [300], carbon nanotubes [301], graphene [302,303], and carbon black [304], further advancements continue to be explored. However, the incorporation of carbon nanostructures onto the surface of CPs remains limited due to challenges including the poor dispersion ability of regular CPs such as PPy and PEDOT, handling of toxic conditions, cost, and production time. Various metal nanostructures, alloys, and metal oxides are gaining prominence in the field of wearable electronics. However, due to concerns related to the cytotoxicity and leaching of heavy metals, liquid metals and various alloys are not suitable for use in the manufacture of wearable electronics and sensors that prioritize human and environmental friendliness [51,124]. CPs can achieve adhesion and robust properties by forming conductive networks on diverse substrates [141].

In wearable sensors, CPs play a dual role as a signal transmission layer and a direct recipient of stimulation signals, distinguishing them from biosensors. In particular, this review highlights the development and importance of CPs and their composites in pressure and strain sensing applications. The use of CPs and their composites could be extended to various fields such as energy production, energy storage, the Internet of Things, artificial intelligence, soft robotics, and artificial limbs. In the field of wearable sensors, durability and flexibility are essential. Capacitive pressure sensors typically consist of up and down electrodes, a dielectric layer film, and a substrate. In comparison with other types of pressure sensor, they consume less power, respond faster, and exhibit less signal drift. Capacitive pressure sensors operational mechanism is presented in Figure 14. External pressure changes the area of the contact surface between the dielectric materials and the electrodes, and the interelectrode distance, resulting in changes in capacitance [305,306]. PANi, PPy, and PEDOT are extensively employed in CPs for sensor applications. Despite their inherent high electrical conductivity, these materials are occasionally paired with supplementary substances such as metals or carbon to enhance their inherent properties, resulting in an improved end product [307]. The four main mechanisms of wearable sensors are all based on the deformation and change in electrochemical parameters in the transducer layer. These mechanisms include piezoelectric, iontronic, capacitive, and piezoresistive mechanisms.

### 5.1. Piezoelectric-Based Device

Piezoelectricity is a popular electrical signal transmission method for wearable electronic devices. In this method, electrical polarization occurs in the material when a mechanical force is applied in a specific direction, generating a voltage difference across the opposite surfaces of the material. Several piezoelectric CPs have been used to develop flexible pressure or strain sensors in wearable devices, and they are more attractive materials for piezoelectric sensors than other materials because of their good mechanical flexibility, cost-effectiveness, biocompatibility, chemical inertness, and good piezoelectric coefficient. The piezoelectric effect has been extensively applied in the development of highly sensitive sensors in wearable electronics, particularly in fast response applications such as body motion detection [309,310]. Su et al. developed a muscle fiber-inspired piezoelectric textile sensor with tunable mechanical properties to monitor physiological signals. The sensors satisfied the need for robust characteristics, biocompatibility, and high stretchability, which are important features for next-generation biosensors in wearable electronics. Dopamine was dispersed into electrospun barium titanate/polyvinylidene fluoride hexafluoropropylene (PVDF) nanofibers to enhance their mechanical strength, surface adhesion, and piezoelectric properties. The resulting sensors demonstrated remarkable sensitivity (3.95 VN^−1^). Furthermore, to assess the long-term stability and durability of the developed sensors, a linear motor operating at 1 Hz was used to perform cyclic loading and release of a load of 5 N. The output voltage after 7400 cycles did not decrease by more than 3%, indicating excellent mechanical durability [311]. Moreover, a real-time muscular monitoring system based on a Ag nanowire and PEDOT:dodecyl sulfate hybrid electrodes showed commendable electrical conductivity, with a hydrophobic surface and a uniform conducting network on a substrate when dried. The muscular monitoring system demonstrated adaptability for use in different areas within the human body and was responsive to human body movements [312]. A chemical combination of PVDF in the form of nanofiber with PPy exhibited a maximum piezoelectric response of 15.56 when used for electrocardiogram measurement [313].

### 5.2. Iontronic-Based Device

Iontronic or electronic ion sensing is a sensor-based mechanism that meets the requirements of device sensitivity and addresses parasitic noise. In this mechanism, a nanoscale ion–electron interface is formed between the electrolyte and the electrode in an electronic ion sensor. The operating mechanism of the sensors is based on the variation in the area between the active material and the electrode with the applied voltage. When voltage is applied, the corresponding reactive ions are attracted to the electrode surface, increasing the contact area and generating extremely high capacitance per unit area. This capacitance is typically 1000 times that of a metal oxide parallel plate capacitor [314,315]. The electronic ion pressure sensor converts the applied force into a change in capacitance by altering the electric double layer (EDL) at the interface of the electrode and the dielectric layer. Ionic gels contain large numbers of positive and negative ions in the space between electrodes. The negative ions are attracted to the positive ions, which results in the formation of two EDLs when the applied voltage is increased. This mechanism has no adverse effect on the human body, minimizing allergic effects on the human skin and eliminating the possibility of causing an electric shock [316]. Notably, the CP materials used in this design are often used in sol-gel or hydrogel forms. However, water loss over time reduces the stability and accuracy of sensor designs, necessitating the use of additional protective shells in iontronic sensor design [141].

Wearable and implantable pressure sensors based on a PEDOT:PSS transducer are used for personal health monitoring, with a low operating voltage and low power consumption. A low-power-consumption electronic ion pressure sensor was presented in which an ionic hydrogel is used as the solid gate medium. The resulting device operates at voltages below 1 V, with power consumption in the range of approximately 10 to 10^3^ μW, while maintaining an adjustable sensitivity in the range of 1.0 to 10 kPa^−1^. The device’s performance was measured under 0 to 200 Pa cyclic pressure, and a stable baseline was recorded with a small change from 2.415 × 10^−3^ to 2.413 × 10^−3^. The stability persisted within 100 operation cycles [305]. Huang et al. introduced aramid nanofibers/CNT/PPy aerogel fibers for use as multifunctional sensors to monitor human health and movement. In their study, lightweight aramid nanofibers (ANFs) and carbon nanotube aerogel fibers coated with PPy layers were prepared by wet spinning for motion detection and information transmission. ANF/CNT/PPy aerogel fibers with a low density of 56.3 mg/cm^3^, conductivity of 6.43 S/m, and tensile strength of 2.88 MPa were used as motion sensors with a high sensitivity of 0.12 and a durable lifespan of 1000 cycles. Furthermore, the differential electrical conductivity of the aerogel fibers was utilized to reduce the information transmission time by 46% at very low temperatures (−196 to 100 °C) [317].

### 5.3. Capacitive-Based Device

Capacitive pressure sensors transform mechanical pressure signals into capacitive signals by manipulating the capacitance. The capacitance value is determined by the formula *C* = ε_0_ε_r_*A*/*d*, where ε_0_ is the vacuum permittivity (8.854 × 10^−12^ F/m), ε_r_ is the relative permittivity, *A* is the active area, and *d* is the distance between adjacent electrodes [318,319]. The active region of this sensor, which is based on mechanical principles, depends on the degree of compressive strain and applied force. The distance between adjacent electrodes (*d*) changes significantly with a change in the pressure level. The capacitance is primarily influenced by the dielectric material’s low elastic modulus, which allows it to deform easily even under gentle pressure. To enhance the sensitivity and response speed of such sensors, microstructured electrodes and optimized dielectric architecture are employed in their fabrication. Tahir et al. introduced a high-performance micro-supercapacitor based on PPy/CNT deposited on an rGO-doped Au substrate. The micro-supercapacitor was fabricated by the electrochemical co-precipitation of PPy/CNT nanostructures on an rGO-doped Au substrate. The resulting micro-supercapacitor demonstrated a high areal capacitance of 65.9 mF/cm^2^ at a current density of 0.1 mA/cm^2^. It exhibited exceptional cycling performance, maintaining 79% of the capacitance after 10,000 charge/discharge cycles at 5 mA/cm^2^. Using intermediate graphene, another study fabricated a current-collector-free PPy/CNT/rGO-doped Au flexible micro-supercapacitor via a facile transfer method on a flexible substrate. The supercapacitor exhibited a capacitance of 70.25 mF/cm^2^ at a current density of 0.1 mA/cm^2^, retaining 46% of the initial capacitance at a current density of 1.0 mA/cm^2^. The flexible micro-supercapacitor was used as a skin-compatible micro-capacitive strain sensor and demonstrated excellent electromechanical properties [320].

In a study by Ferrari et al., a personalized electrode interface implemented as a skin-tattooed system was employed. They used ultrathin and conformal PEDOT:PSS electrodes to capture biological signals from the epidermis. This innovative approach yielded high-quality, artifact-free recordings, surpassing conventional macroscopic electrodes in gel-based state-of-the-art setups. The capacitive coupling phenomenon shows that the substantial values of modulus Z observed at mid-low frequencies (below 102 Hz, a key frequency range in surface electrophysiology) do not impede the inherent capacity of CP electrodes to faithfully transmit biological signals. Tattoo electrodes establish a completely dry and stable connection, and they offer the advantage of maintaining consistent capacitance at the interface even during movement. Consequently, this configuration facilitates precise and artifact-free data collection, which is a notable improvement over contemporary macroscopic and gel-based electrode systems [321]. Furthermore, Li et al. presented a flexible capacitive pressure sensor based on a porous PDMS dielectric layer, with an air gap (through hole) and high porosity (∼60%). PPy-coated filter paper was used as a rough and porous electrode to establish contact with the porous PDMS dielectric layer. The combination of the pore/air gap in the dielectric layer and the rough PPy electrode provided high sensitivity and stable pressure sensing over a wide pressure range. Notably, this system exhibited high sensitivity over the pressure range from approximately 5 Pa to 1 MPa, demonstrating a fast response time and good durability [322].

### 5.4. Piezoresistive-Based Device

Piezoelectric sensors have extensive applications in the measurement of human physiological data because of their straightforward design, ease of use, and user-friendly operation [323]. These sensors exhibit rapid response and recovery times, making them ideal for integration into flexible and wearable devices [324]. The underlying principle of piezoelectric sensors lies in the change in contact resistance (*R*) between two materials when a force is applied. This force is the primary driver of alterations in the electrical signal within the contact area [325]. As strain occurs, a force is exerted on the sensor, causing the contact resistance to vary according to the power law *R*~F^−1/2^. This results in increased sensitivity at lower force levels and an extended operational range. Generally, the relative change in strain sensitivity (Δ*R*/*R*) can be mathematically expressed as *G*_*ε*_ = Δ*R*/*R* = Δ*ρ*/*ρ* + *ε* (1 + 2*υ*), where *ρ* is the resistivity of the transducer material, *R* and Δ*R* are the resistance and the change in resistance, *υ* is Poisson’s ratio, *ε* is the strain degree, and *G*_*ε*_ is the gauge factor. The measurement sensitivity is determined by the magnitude of change in resistivity.

Piezoelectric sensors based on microcracks exhibit exceptional sensitivity to subtle deformations. These microcracks, found within a thin conductive film, alter the contact area of the film, resulting in a significantly higher measurement coefficient. In addition, the use of the tunneling effect further enhances the measurement coefficient, resulting in the sensors having exceptional stretchability. Hence, piezoresistive sensors are primarily employed as wearable pressure and strain sensors, and they have found application in various fields such as motion detection, pulse monitoring, the recognition of sound and vibration signals, tactile sensing, and facilitating human–machine communication [308,326]. Inspired by the properties of human skin, Kong et al. devised a hierarchical structure that effectively addresses the challenge of simultaneously achieving two seemingly contradictory attributes in e-skin: a broad linear range and high sensitivity. They simultaneously achieved both attributes by incorporating rGO and PEDOT:PSS, which involved a piezoelectric mechanism. This innovative structure demonstrated a linear sensing range of up to 30 kPa while preserving high sensitivity at 137.7 kPa−^1^. Notably, it exhibited a rapid response time of approximately 80 ms, an impressive low detection limit of 1.1 Pa, and remarkable stability and reproducibility, enduring more than 10,000 cycles. These features are particularly valuable for applications involving the detection of subtle airflow, monitoring human heart rate, and capturing sound-induced vibrations in electronic skin systems [327].

In a recent development, PPy was deposited onto a cotton fiber layer that served as an electrode in a sensor structure used in a smartwatch. The sensor responded significantly to 10 N pressure interactions and demonstrated long-term stability [328]. PEDOT:PSS, when combined with a soft polymer such as PVA, poly(acrylic acid) (PAA), and PMAA, exhibits noteworthy properties. In a composite based on PAA, PEDOT: PSS exhibited a uniformly dispersed structure, establishing a robust hydrogen bond network between PAA and PEDOT:PSS during blending. This interaction resulted in consistently high and stable electrical conductivity. Moreover, the electrical conductivity of the PEDOT:PSS/PAA composite for a 20/80 weight ratio increased from 13 to 125 S/cm. Furthermore, its stretchability, as indicated by elongation at break, increased from 20% to 40% following methanol treatment. Finally, the methanol-treated PAA-based composite was employed in the development of a pressure-sensing device, and exceptional performance metrics were achieved, including a remarkable sensitivity of 39.90 kPa−^1^ at a 20% strain, a swift response time of approximately 49 ms, and an impressively low detection limit of around 27.4 Pa [329]. In another study, PEDOT:PSS electrodes were fabricated on PDMS molds using a fabric technique. Electrodes were employed in electromyography to monitor the electrical activity of skeletal muscles. The high tolerance of textile electrodes to low-frequency motion artifacts makes them suitable for this application. Electroencephalography is another obvious application for textile electrodes [330]. Shi et al. detailed the fabrication of elastic conductive PPy/PVA films. They prepared the film by adding FeCl_3_/PVA to a solution containing pyrrole and a mixture of acetonitrile and water, allowing in situ polymerization. The resulting strain sensor, fashioned from an optimized membrane, exhibited exceptional stretchability (up to 309.5%), robust mechanical properties (tensile strength of 32.8 MPa), and a notably high relative sensitivity (with the measurement coefficient reaching 5.07 under 1.0% deformation). This sensor is well suited for detecting various subtle physiological signals, including pulse rate and finger joint movements at different frequencies. In addition, it can distinguish between different word pronunciations through vocal cord vibrations [167]. Table 5 outlines the key parameters of CP-based wearable strain sensors for human skin.

### 5.5. Corneal Sensor

Corneal sensors offer a noninvasive yet highly accurate method for directly monitoring biological signals from the body. However, they require strategies to mitigate noise from the environment and blinking activity, often requiring external filter circuits. This results in thickening of the contact lens and a reduction in signal quality. The associated discomfort adversely affects the monitoring accuracy. As shown in Figure 15, Park et al. introduced an innovative stretchable PEDOT doped with tosylate for use in corneal sensors [103]. These sensors were employed to monitor intraocular pressure in patients with glaucoma. They were electrochemically printed onto a disposable soft contact lens mold designed for close and noninvasive contact with the corneal surface of the human eye. The integration of the corneal sensor with the soft contact lens was achieved seamlessly through an electrochemical anchoring mechanism, which guaranteed both mechanical and chemical reliability. This device enabled the precise recording of full-field signals in the human eye, eliminating the need for local anesthesia or speculum anchoring. Notably, electroretinogram signals were successfully recorded for the first time, indicating the sensor’s potential for improvement. The most remarkable advantage of this device was its superior signal quality and enhanced comfort compared with clinical standards, which is a critical consideration for devices in direct contact with sensitive areas such as the eyes [103]. In a previous study, Zhang et al. predicted myopia using PEDOT-functionalized sulfur-doped graphene to fabricate contact lenses capable of measuring dopamine levels in tears. They constructed a high-performance biosensor by electrochemically depositing tyrosinase enzyme, PEDOT, and functionalized sulfur-doped graphene onto a custom-designed corneal microelectrode. This biosensor exhibited remarkable attributes, including a high sensitivity of 12.9 µA × 10^−3^ m^−1^/cm^2^, an LOD of 101 × 10^−9^ m, exceptional selectivity, and long-term stability. In vivo tests on rabbits confirmed a linear relationship between the amount of dopamine added to the animals’ eyes and the biosensor’s current response. Subsequent experiments on tears collected from myopic patients with varying degrees of myopia showed that the biosensor generated a sensitive electrical current signal that was correlated with the severity of myopia diopters. This finding indicates a fundamental connection between the extent of myopia and the dopamine content in tears. This presents a promising avenue for gaining deeper insights into myopia development and devising potential prevention strategies [331].

Ren et al. engineered a highly sensitive and anti-interference contact lens sensor for monitoring intraocular pressure. Achieving precise measurements of this subtle pressure change requires extremely sensitive sensors without amplification while mitigating signal interference caused by eye movements. The contact lens strain sensor consisted of the following key components: a self-lubricating layer formed from a modified PDMS composite with stitch topography, an underlying layer composed of silver nanofibers and a PEDOT:PSS mixture, and a backing layer made of PDMS. This ingenious design incorporated a self-lubricating layer to reduce the coefficient of friction, effectively eliminating interference from tangential friction. What sets this sensor apart is its exceptional sensitivity, achieved through the strategic arrangement of strain concentrations and the deliberate control of symmetrical micro-cracks, which facilitates accurate and interference-free monitoring of intraocular pressure [332].

### 5.6. Artificial Olfactory Sensor

The nose, despite its small size, plays a crucial role in providing humans with a wealth of information. Consequently, the design of tactile sensors requires compact yet elongated dimensions to maximize the electroactive sensing surface, microstructural alignment, and ease of installation [333]. Fang et al. introduced an innovative solvent exchange process for wet-spinning ultrafine PANi fibers, yielding diameters below 5 µm with enhanced energy-storage capabilities. These PANi fibers were employed in ultrathin, all-solid organic electrochemical transistors (OECTs) serving as tactile sensors. The OECT demonstrated excellent amplification efficiency, a high on–off current ratio, and an accurate response to various pressure levels, consuming less than 18 µW of power [334]. Deng et al. developed a bio-mimetic olfactory synapse model integrating an organic electrochemical transistor and a porous solid polymer electrolyte for breath sample analysis. This device exhibited remarkable sensitivity, detecting trace amounts at the parts-per-billion level, and demonstrated high selectivity for hydrogen sulfide. The sensor, designed to emulate the olfactory system, not only possessed sensing capability but also exhibited memory-like functions closely mimicking neurobiological systems [335].

In the research conducted by Hu et al. [336], an in situ growth strategy was employed to establish a repository of gas-sensitive nanocomposites incorporating CP and perovskite. Subsequently, they constructed a system for human identification based on body odor. They fabricated a user-friendly biosensor array module using information from the aforementioned library, which was then integrated into an on-the-go detection platform. Owing to its distinctive Schottky barrier structure, the material library demonstrated enhanced performance, with a 37% to 70% higher response than the original MXene. The intelligent detection platform was used for the precise detection of breath and clothing odors emanating from volunteers, achieving accuracy rates of 69.2% and 51.1%, respectively, with the support of machine learning. To assess the effectiveness of this approach, the researchers developed a comprehensive dynamic modulation method to create a repository of adaptable sensing materials based on MXene and perovskite, thereby establishing a diverse material foundation for the development of a high-performance intelligent detection platform [337]. The development of thin, transparent, and durable self-healing electronic skin (e-skin) has opened up opportunities for multifunctionality in wearable electronics. Nevertheless, the challenge of creating e-skins that can self-repair while maintaining the necessary flexibility, high transparency, and stable operation persists [296,338].

Liu et al. developed an ultrathin e-skin using PEDOT:PSS and a self-healing polymer, exhibiting exceptional transparency and mechanical properties due to amorphous microphase separation. This 3 µm thick e-skin integrates seamlessly with human skin, achieving over 92% transparency in the visible spectrum. It serves as a self-powered sensor, harnessing body motion energy, and maintains consistent electrical output after extensive bending tests and operating cycles. Su et al. presented a stretchable and self-healing CP material with high electrical conductivity, addressing the need for robust materials in wearable electronics. The composite film, combining a CP with a soft polymer, displayed impressive stretchability (up to 630%) and maintained high electrical conductivity (320 S/cm). This versatile polymer composite film is suitable for tactile sensors, exhibiting remarkable pressure sensitivity, an ultrafast response time, and exceptional durability. Integrated OECT arrays enhance its utility in flexible and resilient electronic systems [339,340].

Recent research has also investigated modifications to CP materials, as exemplified in the study of Wang et al. [341], with the objective of achieving high-performance tactile sensors for precise pressure data collection. In their study, a flexible iontronic tactile sensor was developed using conductive PEDOT:PSS thin-film electrodes embellished with polymethyl methacrylate microspheres embedded within an ionic gel electrolyte housed on a polyurethane foam framework. Their investigation focused on the pseudocapacitive sensing mechanism, optimizing both the composition and microstructure of the electrode and polymer electrolyte to achieve remarkably high sensitivity. Their results indicated that the tactile sensor exhibited an impressive sensitivity of 162.90 kPa^−1^, a maximum detection range of 160 kPa, an ultralow detection limit of 30 Pa, a rapid response time of 25 ms, and exceptional stability under pressure loads. The ability to precisely detect subtle pressures associated with minuscule movements has practical applications, including the fabrication and usage of smart gloves equipped with fingertip-mounted sensor arrays. These gloves have been employed for swift and accurate Braille identification. In a complementary effort, Yoon et al. introduced an ultrasensitive and rapid hybrid electronic ion pressure sensor featuring an innovative pressure sensing mechanism and combining a hybrid supercapacitor iontronic membrane, with capacitance stemming from the electrical double layer and pseudo-capacitance. This sensor was constructed by blending carbon nanotubes, PEDOT:PSS, polyacrylamide hydrogel, and vinyl silica nanoparticles. The resulting hybrid supercapacitor pressure sensor had an exceptionally high sensitivity of 301.5 kPa^−1^, covered a wide pressure range of up to 63.3 kPa, and showed a swift recovery time of approximately 32 ms. These advancements substantially reduce response times and widen the operational pressure range of the sensor, making it a promising device for various applications [342]. Table 5 reported the CP-based wearable sensors in previous studies for monitoring human skin.

**Table 5 ijms-25-01564-t005:** Reported parameters of CP-based wearable sensors for monitoring human skin.

Composition/Structure	Target	Sensitivity	Detection Limit	Linear Range	Response Time	Voltage
PPy-PVA co-ethylene) nanofibers [343]	Human skin	1.24 kPa^−1^	1.3 Pa	150 Pa	---	2.0 V
PEDOT:PSS/Nanowires [344]	Human skin	0.97 kPa^−1^	20 Pa	20 to 90 kPa	60 ms	0.1 V
PEDOT:PSS on micropyramid array [325]	Human skin	4.88 kPa^−1^	13 Pa	0.37 to 5.9 kPa	---	0.2 V
PANi foam [345]	Human skin	0.055 kPa^−1^	>4 Pa	0 to 5.0 kPa	90 ms	---
PANi nanofibers/Au-PDMS nanopillars [346]	Human skin	2 kPa^−1^	15 Pa	0.015 to 1.5 kPa	50 ms	1.0 V

## 6. Conclusions: Trends and Challenges

The landscape of sensors based on CPs is undergoing a transformative phase, marked by discernible trends and challenges. Researchers are actively refining the chemical and physical attributes of CPs to elevate sensor performance, with a focus on achieving heightened sensitivity, selectivity, and faster response times. The integration of nanotechnology, particularly through nanocomposite structures involving CPs and nanoparticles or nanotubes, is gaining momentum, offering improvements in conductivity and unique properties.

The trend extends to the development of flexible and wearable CP sensors designed to conform to bio surfaces, finding applications in personalized health monitoring devices. Biosensing applications are expanding, leveraging the biocompatibility of CPs across medical diagnostics, environmental monitoring, and food safety. A noteworthy trend includes the exploration of printable electronics, with a focus on large-scale, cost-effective sensor manufacturing on diverse substrates. In addition, one of the most promising trends in recent times is the development of polymer ionic liquids (PILs), which are polymerized forms of ionic liquids (ILs) or polyelectrolytes that incorporate IL moieties along the polymer backbone. The current fascination with ionic liquids arises from their unique characteristics, including their low vapor pressure, thermal stability, and non-flammability, combined with high ionic conductivity and a broad electrochemical intensity range [347,348,349,350]. These attributes make IL-based electrolytes versatile for various applications, including biosensing and electrochemistry. The commercial potential in the realm of CP investment for human health monitoring sensors is underscored by the development of flexible mobile devices such as foldable phones, smartwatches, and wearable devices.

Despite these promising trends, challenges persist in the field of CP-based sensors. Stability and long-term performance remain areas of concern, necessitating strategic approaches to address potential deterioration over time. Achieving reproducibility in sensor fabrication poses challenges due to variability in synthesis methods and environmental factors. Ensuring a high selectivity, particularly in complex sample matrices, remains an obstacle, requiring enhanced discrimination between analytes and potential interferences. Scalability for commercial applications is another challenge, demanding the translation of promising laboratory-scale results to larger production scales without compromising performance. Furthermore, a comprehensive understanding of the fundamental mechanisms governing CP interactions in a sensor is crucial to enhance sensor design and performance. As the field navigates these trends and challenges, continuous advancements are anticipated in the application of conductive polymer sensors across diverse domains.

## Figures and Tables

**Figure 1 ijms-25-01564-f001:**
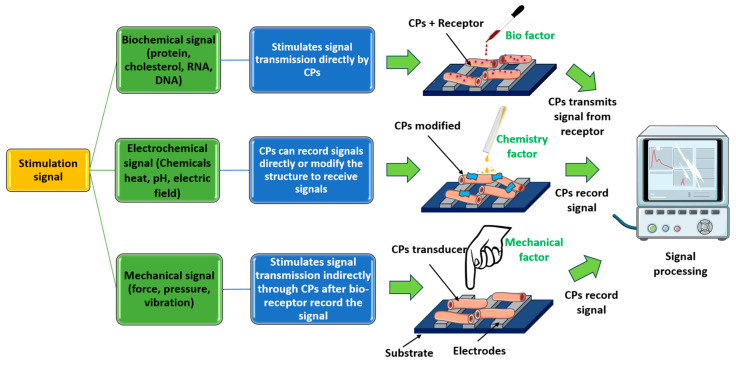
Role of CPs in receiving and processing stimulation signals.

**Figure 2 ijms-25-01564-f002:**
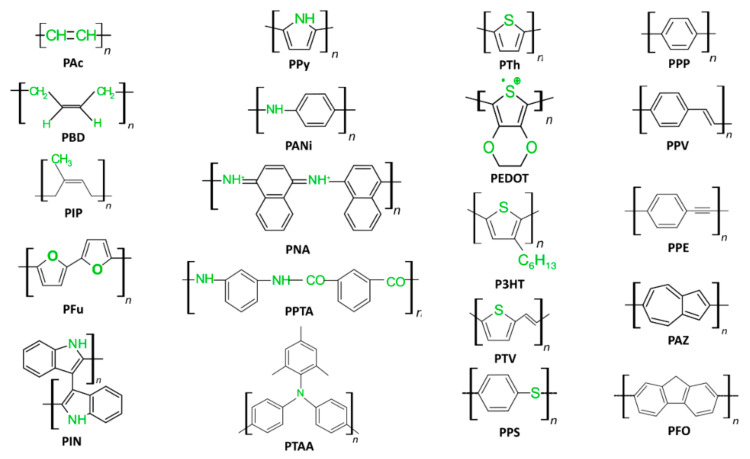
Inventory of CPs along with their abbreviations.

**Figure 3 ijms-25-01564-f003:**
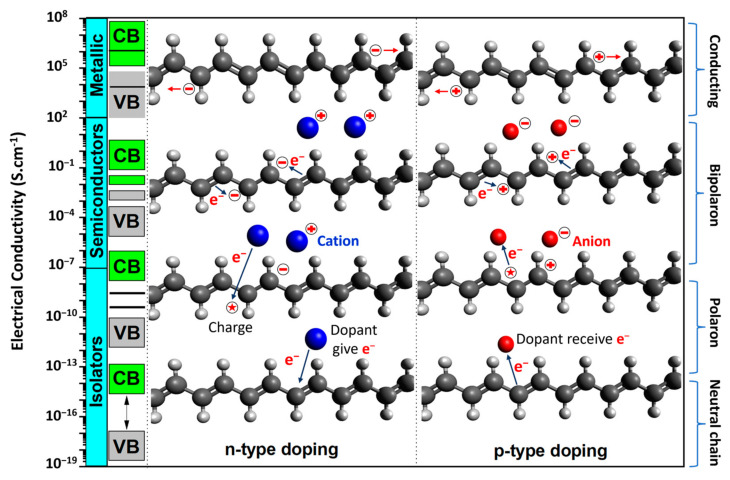
Conduction mechanism of PAc, a typical of CPs. (Referenced from Ref. [8] Copyright (2018), with permission from MDPI Publisher).

**Figure 5 ijms-25-01564-f005:**
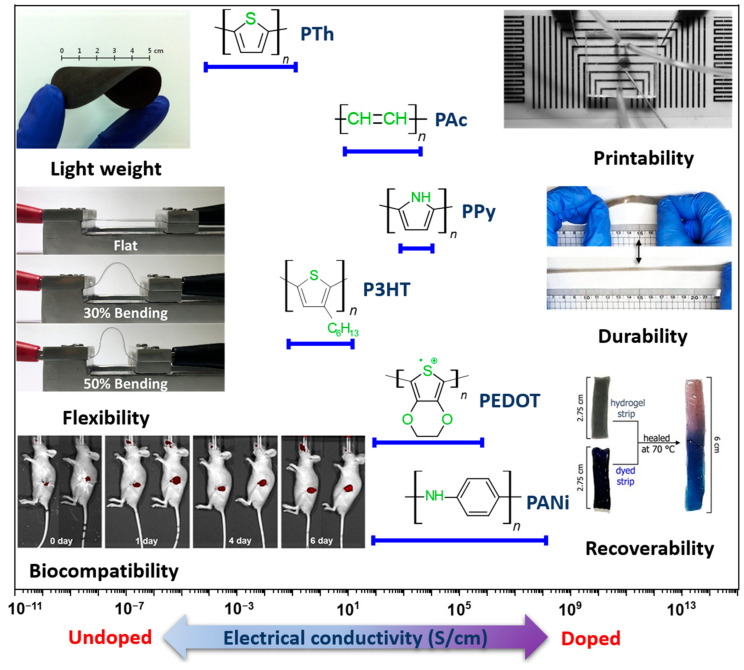
Advantages of using CPs for biosensors (Reprint from Ref. [121] Copyright (2018), with permission from Elsevier Publisher). (Reprint from Ref. [122] Copyright (2015), with permission from Springer Nature Publisher). (Reprint from Ref. [123] Copyright (2020), with permission from ACS Publisher). (Reprint from Ref. [124] Copyright (2020), with permission from MDPI Publisher).

**Figure 6 ijms-25-01564-f006:**
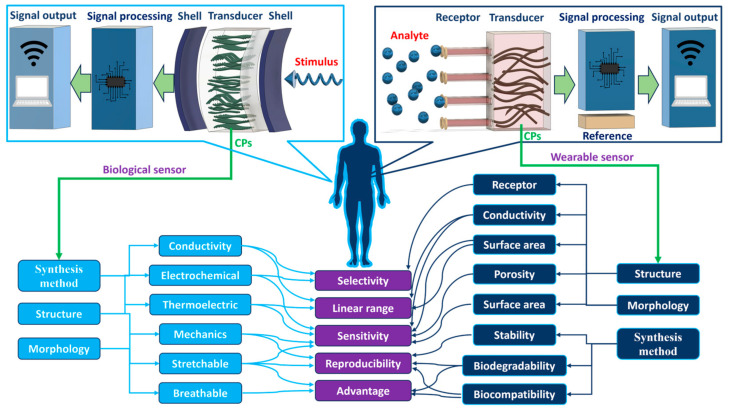
Main properties and roles of CPs in human health sensors [141,142,143].

**Figure 7 ijms-25-01564-f007:**
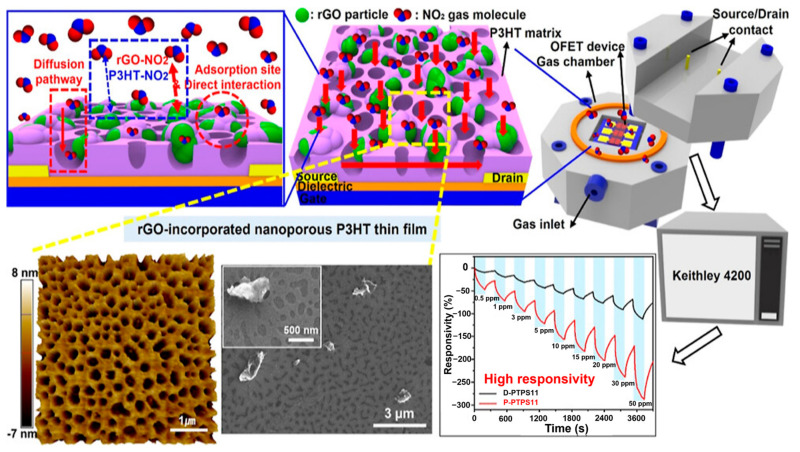
Schematic of the NO_2_ sensing performance of an OFET based on CPs; composite of rGO-incorporated nano-porous films. (Reprint from Ref. [213] (copyright (2023) ACS Publisher).

**Figure 8 ijms-25-01564-f008:**
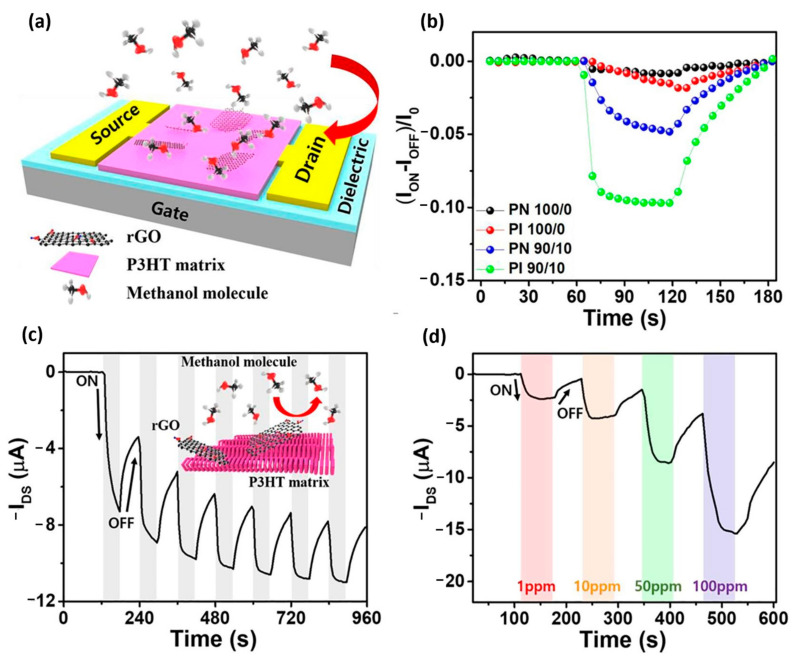
Design of OFET sensor based on CPs. (**a**) the photo-irradiated P3HT and rGO composite films were used in the OFET configuration. (**b**) Comparison of responses of OFET sensors based on pristine P3HT, pristine P3HT/rGO (90/10), photo-irradiated bare P3HT, and photo-irradiated P3HT/rGO (90/10) to 10 ppm methanol vapor. (**c**) Photo-irradiated P3HT/rGO showed consistent methanol vapor sensing (90/10). (**d**) The composite depicts the interactions between methanol vapor and rGO molecules, and it shows how a photo-irradiated P3HT/rGO (90/10) sensor responds to different methanol vapor concentrations. Testing of the OFET sensor was performed in a continuous environment [227]. (Reprint from Ref. [227] (copyright (2021) Elsevier Publisher).

**Figure 9 ijms-25-01564-f009:**
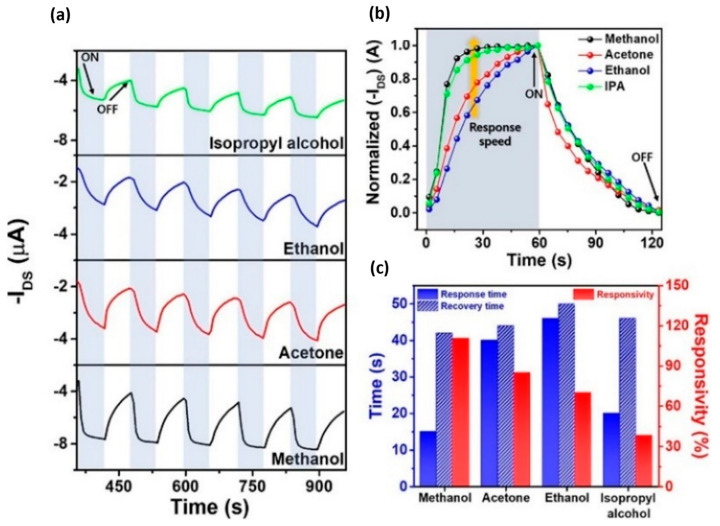
OFET sensor’s ability to identify VOC vapors based on CPs. After exposure to 10 ppm of polar VOC vapors, namely methanol, acetone, ethanol, and isopropyl alcohol, photo-irradiated P3HT/rGO (90/10)-based OFET sensors showed the responses and normalized drain currents displayed in (**a**,**b**). (**c**) Corresponding responsivity (right axis) and response/recovery time (left axis) for 60-s on/off pulses. V_GS_ = −10 V and V_DS_ = −40 V were the constant voltages used for the OFET sensor test [227].(Reprint from Ref. [227] (copyright (2021) with permission from Elsevier Publisher).

**Figure 10 ijms-25-01564-f010:**
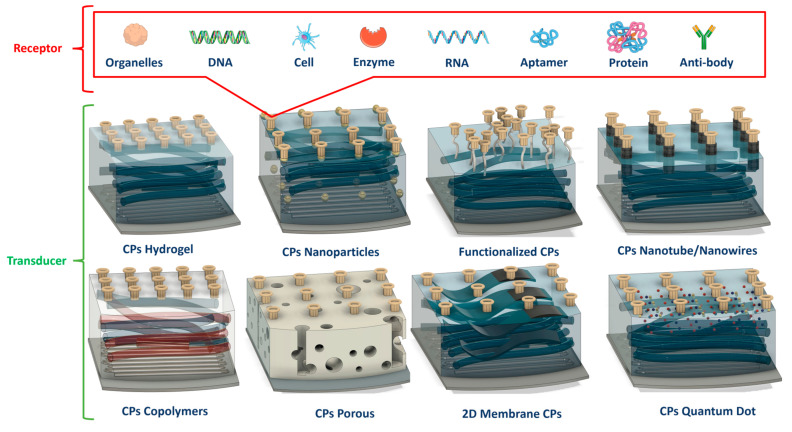
Sensing mechanisms of CPs-based biosensors.

**Figure 11 ijms-25-01564-f011:**
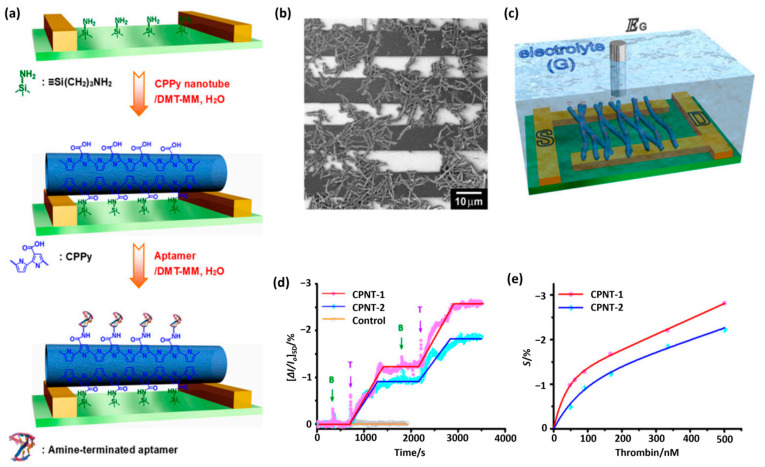
Three different images of CPs nanotube-based aptamer sensor platforms: (**a**) a schematic of the reaction steps, (**b**) an FE-SEM image of a typical CPPy nanotube deposited on an interdigitated microelectrode substrate, (**c**) a schematic of a CPPy nanotube sensor platform with the liquid-ion gate (G), drain (D), and source (S) marked. (**d**) The CPPy nanotube FET sensors’s real-time responses were measured at VSD = −15 mV for 1 CA: 15 PPy (CPNT-1) and −10 mV for 1 CA: 30 PPy (CPNT-2). The ISD changed upon the addition of 90 nM target (thrombin, T) and nontarget (BSA, B) proteins in succession (the arrow denotes the addition of protein solutions). (**e**) The calibration curves show that the sensitivity of the CPPy nanotube FET sensors changed with the thrombin concentration. (Reprint from Ref. [291] Copyright (2008), with permission from Wiley Publisher).

**Figure 12 ijms-25-01564-f012:**
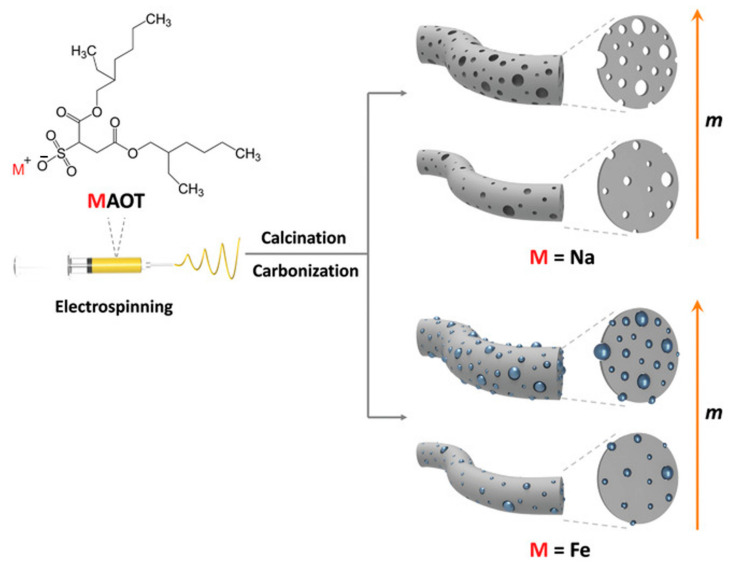
Overview of the modified porous fibrous process with precisely regulated substructures by using surfactant-in-polymer templating, applied for CPs. (Reprint from Ref. [293] Copyright (2021), with permission from Wiley Publisher).

**Figure 13 ijms-25-01564-f013:**
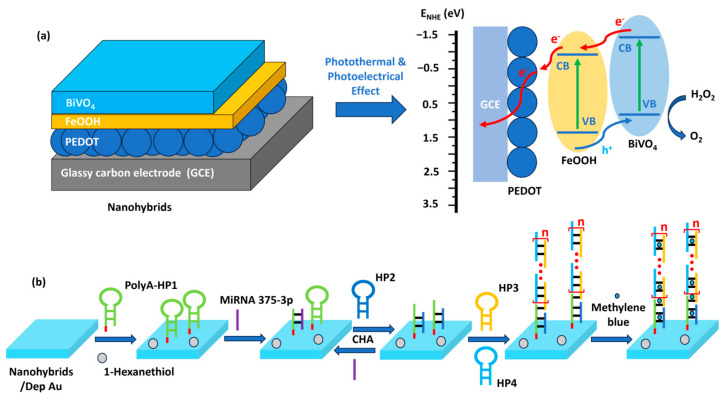
Schematic diagram of the biosensor based on positive electrodes from CPs. (**a**) Positive electrode structure with PEDOT transducer layer excited by photothermal and photovoltaic effects. (**b**) PEC biosensor assembly process. (Redraw from Ref. [297] Copyright (2021), with permission from ACS Publisher).

**Figure 14 ijms-25-01564-f014:**
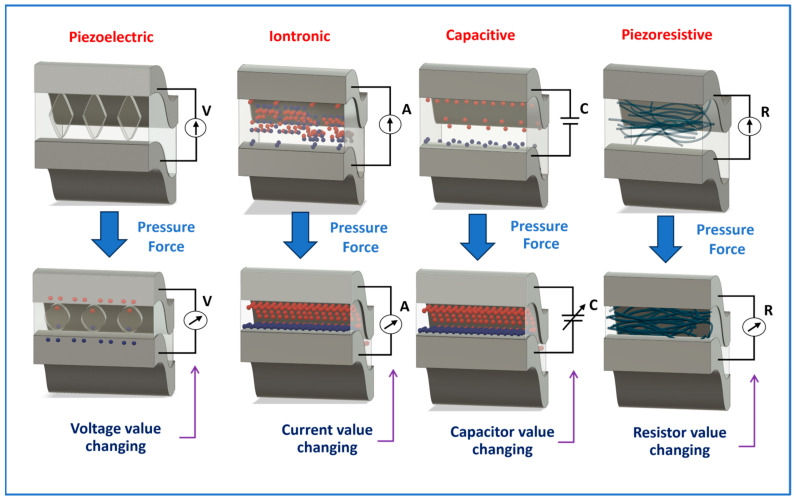
Sensing mechanisms of CP-based wearable sensors (Referenced from Ref. [308] Copyright (2022), with permission from MDPI Publisher).

**Figure 15 ijms-25-01564-f015:**
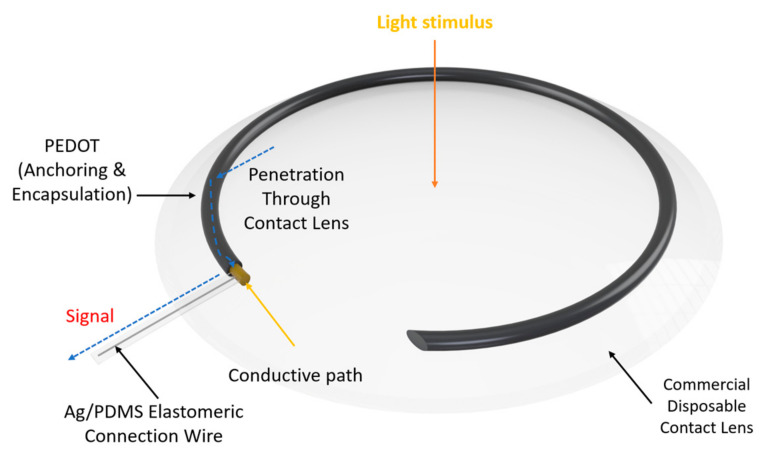
The design of a corneal sensor based on CPs. (Referenced from Ref. [103] Copyright (2021), with permission from Springer Nature Publisher).

**Table 1 ijms-25-01564-t001:** Comparison of different CP polymerization methods.

Method	Advantages	Disadvantages
Chemical	Scalable production, easy to functionalize, involves chemical doping, straightforward and efficient route, yields a composite material, easy to control the morphology	Time-consuming, only thick films and powders can be synthesized, CPs may impurify, heterogeneity optimization, indirect approach
Electrochemical	No oxidizing agent is required, requires less time, produced CPs have high electrical conductivity, thin films can be produced with dimensional control. Eco-friendly, solventless, efficient, requires a small amount of chemical	Large-scale production is not possible, it is difficult to remove the grown film from the electrode surface, larger reactors are required
Vapor-phase polymerization	Covers many surfaces, monomer easily combines with volatile substances, no solvent required, few impurities	Long duration, low yield, hard to control synthesis, CPs heterogeneity

**Table 3 ijms-25-01564-t003:** Effect of CP polymerization and morphology on sensor performance.

Composition	CP Polymerization	CP Morphology	Target and LOD	Linear Range
PAc-chitosan [238]	Electrochemical	Thin film	Methyl parathion 2.0 × 10^−9^ mol/L	2.0 × 10^−8^ to 1.0 × 10^−4^ mol/L
PAc-CeO_2_ [239]	Vapor phase	Nanohybrids	Megestrol acetate 1.30 nM	4.20 × 10^−8^ to 1.13 × 10^−6^ M
PPy [240]	Chemical oxidate	Thin film	Interleukin-10 (0.347 pg/mL)	1.0 to 10 ng/L
Pt-PPy [241]	Chemical oxidate	Nano-spherical	Hg^+^ 0.277 nM	5 to 500 nM
PPy [242]	Electrochemical	Soft globular	Carcinogenic embryonic antigen 0.13 pg/mL	0.125 to 12.25 pg/mL
PPy-pTS [243]	Electrochemical	Thin film	Hypoxanthine 5.0 µM	5.0 µM to 5.0 mM
PPy-pyrrole [244]	Electrochemical	Macroporous	Urea 2.57 mM	1.67 to 8.32 mM
PPy-MWCNTs [245]	Electrochemical	Solgel	Tramadol 0.03 nM	0.2 to 20 nM
P3HT-PEDOT:PSS [246]	Electrochemical	Thin film	NH_3_ and humidity ---	2.0 to 100 ppm, 10% to 60% RH
P3HT-*b*-(poly(3-triethylene-glycol-thiophene) [247]	Chemical oxidate	Nanoparticle core–shell	*E. coli* BL21 500 cfu/mL	10^3^ to 10^7^ cfu/mL
P3HT-electrolyte-gated organic field-effect transistor [248]	Electrochemical	Nanofibers	Streptavidin 1.47 nM	1.6 × 10^−3^ to 1.6 μM
Au-PANi [249]	Chemical oxidate	Nanoparticles	Melamine 1.39 pM	10 to 10 mM
PANi-GO [250]	Chemical oxidate	Nanocomposite	DNA 20.8 fM	10^−15^ to 10^−6^ M
PANi-PSS [251]	Electrochemical	Macroporous	Alpha-fetoprotein 3.7 fg/mL	1000 pg/mL
PANi-anti-*E. coli* antibody [252]	Electrochemical	Thin film	*E. coli* 10 cfu/mL	10 to 10^6^ cfu/mL
PANi [253]	Electrochemical	Thin film	Urea ---	10^−5^ to 0.1 M
SnO_2_-3D rGO-PANi [254]	Electrochemical	Thin film	Glucose 0.26 nM	0.1 to 5.0 μg/mL
Graphene–copper phthalocyanine and PANi [255]	Electrochemical	Nanolayer	Glucose 6.3 × 10^−8^ M	5 × 10^−7^ to 1.2 × 10^−5^ M
PEDOT-hydroxyapatite [256]	Electrochemical	Thin film	Nitrite 83 nM	0.25 to 1050 µM
PEDOT [257]	Electrochemical	Thin film	Cancer gene BRCA 1 0.0034 pM	0.01 pM to 1 nM
PEDOT-COOH [258]	Electrochemical	Nanofibers	Lactate ---	20 to 960 μM
PPy-PEDOT-Ag [259]	Chemical oxidate	Nanocomposite	DNA 5.4 fM	0.01 to 10 pM
g-C_3_N_4_-PEDOT [260]	Vapor phase	Nanofiber	Acetaminophen 34.28 nM	0.01 to 2 mM
Chitosan-PTh-CdTe [261]	Chemical oxidate	Thin film	Carcinoembryonic antigen 40 fg/mL	10^−4^ to 10^4^ ng/mL
PTh-methylene blue [262]	Electrochemical	Thin film	Albumin 3.0 × 10^−11^ g/L	1.0 × 10^−10^ to 10^−4^ g/L
Epoxy-substituted PTh [263]	Chemical oxidate	Thin film	IL-1α cancer biomarker 3.4 fg/mL	0.01 to 5.5 pg/mL

**Table 4 ijms-25-01564-t004:** Report of CPs transducer sensor design and performance.

Target Analytes	Transducer Design	Linear Range	LOD	Signal Reorganization
Phenanthrene	PANi-WO_3_/Gr [277]	1.0 to 6.0 pM	0.123 pM	Cyclic voltammetry
4-Aminophenol	Gr-PANi [278]	0.2 to 20 mM	6.5 × 10^−8^ M	Differential pulse voltammetry
Melamine	Au-PANi [249]	10^−6^ to 10 μM	1.39 pM	Differential pulse voltammetry
Cd^2+^	PANI-MWCNTs [279]	0.02 to 20 μg/L	0.01 μg/L	Anodic stripping voltammetry
Zn^2+^	Nafion-GR-PANi [280]	0.0 to 0.1 μg/L	1–300 μg/L	Square wave voltammetry
Dopamine	Gr-PANi [281]	0.007 to 90 nM	0.00198 nM	Square wave voltammetry
H_2_O_2_	PEDOT-Prusian Blue [282]	0.5 to 839 μM	0.16 μM	Amperometric *i*–*t* curve
Dopamine	PEDOT-rGO [283]	0.1 to 175 µM	39 nM	Amperometric *i*–*t* curve
Vitamin B2	PEDOT-ZrO_2_NPs [284]	1.0 to 300 μM	0.012 μM	Differential pulse voltammetry
Acetaminophen	PEDOT–GO [285]	10 to 60 μM	0.57 μM	Cyclic voltammetry
Catechol	PEDOT–rGO-Fe_2_O_3_-PPO [286]	0.04 to 62 μM	7.0 nM	Differential pulse voltammetry
Bisphenol A	PPy-GrQDs [287]	0.1 to 50 μM	0.04 μM	Differential pulse voltammetry
H_2_O_2_	Mn/PPy NWs [288]	5.0 to 90 μM	2.12 μM	Amperometric *i*–*t* curve
Amaranth Ponceau 4R	CNTs-PPy [289]	5.0 to 500 nM	0.5 nM	Square wave voltammetry
Acetaminophen	PPy-GO [284]	10 to 60 µM	0.57 µM	Cyclic voltammetry
Nitrate	PdNCs-PPy [290]	0.1 to 1.4 mM	0.74 μM	Differential pulse voltammetry
Methyl parathion	PAc-chitosan [238]	2.0 × 10^−8^ to 1.0 × 10^−4^ mol/L	2.0 × 10^−9^ mol/L	Cyclic voltammetry
Carcinoembryonic Antigen	Chitosan-PTh-CdTe [261]	10^−4^ to 10^4^ ng/mL	40 fg/mL	Cyclic voltammetry
Albumin	PTh-methylene blue [262]	1.0 × 10^−10^ to 1.0 × 10^−4^ g/L	3.0 × 10^−11^ g/L	Cyclic voltammetry
IL-1α cancer biomarker	Epoxy-substituted PTh [263]	0.01 to 5.5 pg/mL	3.4 fg/mL	Cyclic voltammetry

## Data Availability

Data contained within the article.

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
