# Peer review of "Advances in the Use of Conducting Polymers for Healthcare Monitoring"

_ijms, 2024, doi:10.3390/ijms25031564_

Round 1
Reviewer 1 Report
Comments and Suggestions for Authors
This extensive review by Cuong Van Le and Hyeonseok Yoon promises an an overview of the use of conducting polymers for healthcare sensing. The authors have collected an impressive amount of literature on the subject, but the review itself is indigestible as it lacks the context, full of unnecessary information that only inflates the volume and the number of references, is ill-structured, and lacks analysis. The Abstract and the Conclusion contain nothing but empty general statements.
The motivation for the use of sensors sounds strange: ". Sensory 32 receptors located in the nose, eyes, ears, skin, and specific regions of the body predomi- 33 nantly capture these stimulating signals. Specifically, the human body possesses mecha- 34 nisms to interpret these signals. However, instances of distortion may arise ... This challenge is a primary motivator for advancing the development of more precise 39 and efficient sensor designs." So, inn the absence of noise and distortion sensors are superficial and one can rely fully on body reactions?
Then the review opens with long lists of conducting polymers and sensors in which they were used, which are meaningless at thhis stage, though immediately giving the first hundred of references. Then, one does not need a hasty introduction in electronic structure, conductivity mechanisms, and synthesis of conducting polymers here: there are excellent reviews available, sufficient to refer the reader to a few of these.
What one needs is a structured introduction to the field, something like: what does one need to monitor? which signals and how are registered? which biosensors exist on the market? why and how conducting polymers can improve on these?
No sense to go from polymer to polymer giving examples in which sensors each of them was used. By the way, polyacetylene is NOT acetylene black!
In the following, not much sense just giving a summary of "who did what": one rather needs analysis, how good or bad these results were.
In all, the knowledge of the literature the authors have acquired sould be digested to help the reader better understand what are the achievements and challenges of this field.
Comments on the Quality of English LanguageLanguage is OK.
Author Response
Dear Reviewer,
We are submitting a revised version of the manuscript (ijms-2807987) entitled “Advances in the use of conducting polymers for healthcare monitoring” for publication as a full paper in Molecular Sciences journal.
We highly appreciate your and the reviewers’ positive and valuable comments on our manuscript. We have carefully revised the manuscript according to the reviewers’ comments. Please see the attachment below.
We believe that the manuscript has been improved by considering the reviewers’ comments. Once again, we greatly appreciate the reviewers’ thoughtful comments that helped improve the quality of the manuscript.
Thank you for your consideration. I am looking forward to hearing from you. Please do not hesitate to contact me if any additional information is needed.
With best regards,
Prof. Hyeonseok Yoon
School of Polymer Science and Engineering,
Chonnam National University, Gwangju, South Korea

Reviewer 2 Report
Comments and Suggestions for Authors
The review focuses on the latest developments in the field of conductive polymers in relation to healthcare monitoring. The idea of the manuscript's theme is very good, the authors have studied a lot of thematic literature and presented their views on this important field of application of conductive polymers well. The review can be accepted for publication after correcting the following comments.
The authors' work is mainly focused on the problems and breadth of applications of conductive polymers in the biomedical field, therefore it should be interesting for readers, and therefore there should be an easy language and changes made throughout the review in order to attract and improve the understanding of a wide range of scientific readers.
1. It is necessary, if possible, to divide huge (and therefore difficult to read) paragraphs into 2-3 logical paragraphs of smaller volume. For example, a paragraph (Lines 75-113) can be divided by sentence in Line 96, especially further along the text, the authors set tasks and goals for the entire review (which indicates good command of the material and clarity of presentation). Or Section 5.4 consists of just one huge paragraph. Or the first paragraph of Section 5.6. This comment can be applied to the entire text of the manuscript.
2. Despite the fact that there are 12 figures and 5 tables, each subsection should be additionally equipped with its own thematic figure (for example, Section 4.2 would be better if it contained a figure and other informative sections too).
3. Lines 62-74. The authors are recommended to provide a table of abbreviations for all CPs in the review.
4. The authors have practically left the polymeric ionic liquids outside the scope of their consideration. And this is one of the most promising classes of compounds in the field under consideration! Ref 184 is very good, but this is a 2004 work. It is strongly recommended, for example, to mention modern work (http://dx.doi.org/10.1039/D2BM00046F ) and a very recent review (http://dx.doi.org/10.1039/d3gc02131a ).
5. Section 2.2.2 Lines 212-215. The phrase is quite confusing. In addition to constant voltage (potential), CPs synthesis can also be carried out at a controlled current too. It is recommended to give the types of mechanisms in the form of a scheme. Figure 3 shows a two-electrode device for electrochemical polymerization, and the text describes a three-electrode circuit.
6. Sections 5.1-5.4 use the term Mechanism in their title, which can be interpreted ambiguously. Perhaps the term Device is more appropriate for work focused on various CPs applications.
7. Captions to the figures must contain permission for publication.
8. The references provided by the authors are quite modern, but the year 2023 is coming to an end, and novel interesting studies have already appeared. It is recommended to add of 5-7 more works of the most recent publications, the choice is at the discretion of the authors, who could emphasize the trends and advantages of CPs for biomedical purposes.
9. The list of references should contain the correct abbreviations of journal titles.
Author Response

(The authors gave the same response as above.)

Round 2
Reviewer 1 Report
Comments and Suggestions for Authors I recommend acceptance and publication of this work in its present formAuthor Response
Dear Reviewer
We appreciate your thorough review of the manuscript. Your insightful comments have been invaluable in shaping the final version of the work. We are pleased to hear that you recommend acceptance and publication in its present form. Your endorsement reinforces the efforts put into this research, and We are grateful for your positive assessment.
Thank you for your time and consideration.
Best regards,
Reviewer 2 Report
Comments and Suggestions for Authors
Prof. Hyeonseok Yoon and other authors have done a great job to improve their manuscript, which has been greatly transformed in a positive way, and the review will soon be close to acceptance in IJMS. There are only some unclear points that I would like to get more information on, and there are also possible the corrections to the comments already made.
5. Section 2.2.2 Lines 212-215. The phrase is quite confusing. In addition to constant voltage (potential), CPs synthesis can also be carried out at a controlled current too. It is recommended to give the types of mechanisms in the form of a scheme. Figure 3 shows a two-electrode device for electrochemical polymerization, and the text describes a three-electrode circuit.
The comment has been partially corrected. Now these are Lines 233-236. You may need to correct an obscure phrase as follows:
“…Applying a voltage to these electrodes triggers a redox reaction that results in the formation of the polymer. Electrochemical polymerization can be further categorized into cyclic voltammetry and potentiation methods, depending on whether a cyclic or a constant voltage is applied… Electrochemical polymerization can be further categorized into the following methods. Electrochemical synthesis of CPs can be carried out both at a controlled current and at a controlled potential of the working electrode. Also, using the method of cyclic voltammetry, CPs can be synthesized already at a potential (voltage) that changes during cycling. "
And the response to Comment 4 contains a corrected fragment (for the Page 39, 2nd paragraph), which is slightly different from the correction in the manuscript. I suppose both options are quite correct, but still, which one is the final option? I ask the authors to decide on this.
Author Response
Dear Reviewer,
We are submitting a revised version (round 2) of the manuscript (ijms-2807987) entitled “Advances in the use of conducting polymers for healthcare monitoring” for publication as a full paper in Molecular Sciences journal.
We highly appreciate your and the reviewers’ positive and valuable comments on our manuscript. We have carefully revised the manuscript according to the reviewers’ comments. Please see attach file below.
Thank you for your time and consideration.
Best regards,
Round 3
Reviewer 2 Report
Comments and Suggestions for Authors
The manuscript is ready for acceptance in its current form.